# Telomere-to-telomere genome assemblies and population resequencing of diploid and allotetraploid peanut varieties

Jianxin Bian[1,7], Yilin Zhang [1,2,7], Shuai Ding[1,7], Haosong Guo[1,3,7], Kui Li[1], Yu Guan[1], Guoliang Zhou[1], Jihua Li[1], Vanika Garg [3], Yuanyuan Cui[1], Yuan Lv[1], Annapurna Chitikineni [3], Qingjing Meng[1], Tianyu Li[1], Liangqiong He[4], Chuanzhi Zhao [5], Xingjun Wang [5], Ronghua Tang[4], Liangsheng Zhang [6], Xing Wang Deng [1,2], Rajeev K. Varshney [3] ✉, Hang He [1,2] ✉ & Xiaoqin Liu [1] ✉

Peanut (*Arachis hypogaea* L.) is a globally significant leguminous oil crop. Here we present telomere-to-telomere genome assemblies for two diploid and four tetraploid peanut varieties, resulting in high-quality reference genomes, showing that the complex activities of transposable elements, chromosomal rearrangements and centromere expansions within subgenomes collectively contribute to the asymmetrical evolution of the tetraploid genome, and unique structural variants in the four tetraploid peanut varieties provide clear evidence of domestication. Population analyses of 521 peanut accessions revealed asymmetric selection events between subgenomes during breeding, and genome-wide association studies identified candidate genes linked to oil content, seed size and weight, kernel dehydration rate, and arachidic acid content. In addition, transcriptomic and metabolomic analyses revealed enhanced activity in lipidomic and anthocyanin biosynthetic pathways during seed development. These comprehensive findings provide insights into genome organization, evolutionary dynamics and phenotypic differentiation across peanut varieties that could inform future peanut breeding and improvement strategies.

Peanut (*Arachis hypogaea* L.), an important food crop and source of high-quality edible oil, originated in South America. It spread globally in the 16th century and thrives in various climates owing to its adaptability and high nutritional value[1,2]. The genus *Arachis* in the Fabaceae family comprises more than 82 species, including three wild aneuploids ($2n = 2x = 18$), 26 wild diploids ($2n = 2x = 20$), one wild tetraploid (*Arachis monticola*, $2n = 4x = 40$) and one cultivated tetraploid (*A. hypogaea*, $2n = 4x = 40$)[3,4]. *A. hypogaea* (AABB), the only domesticated species, likely originated from the hybridization of two diploids,

*Arachis duranensis* (AA) and *Arachis ipaensis* (BB). It is categorized into two subspecies (var. *hypogaea* and var. *fastigiata*) and six botanical varieties (var. *hypogaea*, var. *hirsuta*, var. *fastigiata*, var. *vulgaris*, var. *aequatoriana* and var. *peruviana*) based on morphology and geography[5–8].

To date, five wild diploid ancestral species[9–12], three wild tetraploid species[11,13] and seven cultivated tetraploid peanut genomes[1,2,11,14,15] have been sequenced and assembled. However, many gaps and unmapped scaffolds remain (with the exception of Yuanza9102), hindering the study of structural variants (SVs) and centromeric evolution.

**Fig. 1 | Phenotypes and genome characterization. a–f,** From top to bottom: plant structures, pods, seeds and Circos plot depicting the genomic features of the V14167 (**a**), K30076 (**b**), S245 (**c**), HN873 (**d**), HN51 (**e**), and S83 (**f**) accessions. The scale for the chromosomes (outer bars) is Mb. The triangles represent telomeres, and the bars represent centromeres. From outside to inside, the different tracks of the Circos plots represent GC content, gene density, TE density, and distributions of *Gypsy, Copia* and *DNA* TEs. The gene and TE densities were calculated based on a 200-kb window size.

The published genomes include those of var. *hypogaea* (Tifrunner, NDH108, ZP06, Mh8)[2,11] and var. *vulgaris* (Shitouqi, Fuhuasheng and Yuanza9102)[1,14,15], whereas var. *hirsuta* and var. *fastigiata* lack genomic sequences. Construction of telomere-to-telomere (T2T) genomes for peanut accessions will enhance identification of SVs and provide a better understanding of genome organization and domestication.

Selection and cultivation of peanut (*A. hypogaea*) have produced diverse varieties with respect to seed size, oil content, pod shape and color, as well as differences in growth habit, disease resistance and maturity time (https://www.ars-grin.gov). Although several quantitative trait loci (QTLs) and genes related to pod and seed size and/or weight[16], oil content[17], branching habit[18] and testa color[19] have been identified using QTL mapping, the limited number of genetic markers has made it challenging to identify causal genes. With advances in sequencing technologies, multiple key loci and genes associated with various traits have been identified[11,20,21].

Here we used two diploid progenitors and four representative allotetraploid peanuts to construct T2T genomes. These six reference genomes, combined with resequencing data from 521 peanut accessions, offer insights into the genetic characteristics and genomic evolution of peanuts, facilitating identification and utilization of superior genotypes for molecular breeding.

## Results

### Assembly and annotation of six representative peanut genomes

To determine the genome structure and genetic diversity of *A. hypogaea*, we sequenced six representative accessions: two diploid progenitors (*A. duranensis*, V14167; *A. ipaensis*, K30076), and four allotetraploid cultivars (Laiyang Silihong, S245; Chedouzi, HN873; Huayu23, HN51; Yunnan Rainbow Peanut, S83) (Fig. 1). These accessions exhibited diverse phenotypes, including differences in tiller angle, flower characteristics, and seed size, shape and number (Fig. 1). Heterozygosity rates ranged from 0.14% to 0.36% (Supplementary Table 1). Using PacBio sequencing, we generated 1.42 Tb (average depth of 116×) of HiFi reads (Supplementary Table 2), assembling genomes with an average contig N50 of 85.94 Mb using Hifiasm (v.0.19.5)[22]

(Table 1). High-throughput chromatin conformation capture (Hi-C) data (average depth of 112×) were used to corrected and orient contigs (Supplementary Fig. 1 and Supplementary Tables 2 and 3), and Oxford Nanopore Technologies (ONT) ultralong-read sequencing was used to fill gaps (average depth of 70×; Supplementary Tables 2 and 4), resulting in T2T peanut genomes ranging from 1.18 Gb to 2.63 Gb (Table 1). Telomeres were identified using 7-bp telomere repeats (CCCATTT at the 5′ end and TTTAGGG at the 3′ end). Assembly quality was assessed using metrics including BUSCO[23] completeness score of 98.67% and long terminal repeat (LTR) assembly index (LAI)[24] score of 21.76, indicating high completeness (Table 1). The accuracy was validated by a quality value[25] of 47.52 and a mapping rate of 99.76% for Illumina short reads (Table 1 and Supplementary Table 5). In addition, *k*-mer distribution curves demonstrated high completeness in the duplicated regions (Supplementary Fig. 2). Transcriptome sequencing achieved a mapping rate of 92.93% (Supplementary Table 6). Our T2T genomes demonstrated superior quality and completeness compared to previously published genomes (Supplementary Fig. 3). Using ab initio, homology and transcript-based evidence, we predicted 34,406 to 75,143 protein-coding genes across the accessions, with an average functional annotation rate of 98.75% (Table 1 and Supplementary Table 7). The average gene length was 2,654 bp, with coding sequences (CDSs) between 1,080 bp and 1,123 bp in length (Supplementary Table 8). BUSCO evaluation showed that 98.88% of single-copy genes were completely annotated, further confirming the accuracy of gene annotations (Table 1).

### TEs and centromeres imply asymmetry evolution

Transposable elements (TEs) are crucial sources of lineage-specific genomic innovation and have vital roles in peanut genome evolution. TEs accounted for an average of 76.27% of each genome, ranging from 74.65% to 77.69% (Table 1 and Supplementary Table 9). The TE content in the Bt subgenomes (average 77.77%) was consistently higher than that in the At subgenomes (average 75.08%), and a similar difference was also observed between the B (77.69%) and A (74.65%) diploid genomes. Specifically, *Gypsy* elements constituted 52.40% (~605.50 Mb) of the A(t) genome/subgenomes, whereas they accounted for 58.91% of the B(t) genome/subgenomes (~865.57 Mb) (Supplementary Table 9). Analysis

**Table 1 | Overview of assembly and annotation statistics for diploid and tetraploid peanuts**

| Sample | V14167 | K30076 | S245 | HN873 | HN51 | S83 |
|---|---|---|---|---|---|---|
| Number of contigs | 405 | 476 | 1,902 | 932 | 1,281 | 1,762 |
| Contig N50 (Mb) | 65.37 | 68.95 | 83.61 | 89.99 | 84.96 | 122.78 |
| Telomere | 20 | 20 | 40 | 40 | 40 | 40 |
| Assembly length (Mb)(gaps) | 1,178(0) | 1,485(0) | 2,572(0) | 2,622(0) | 2,625(0) | 2,620(0) |
| Illumina read-mapping rate (%) | 99.35 | 100 | 99.99 | 99.65 | 99.75 | 99.81 |
| Genome BUSCOs (%) | 97.90 | 99.00 | 98.30 | 98.80 | 98.80 | 99.20 |
| Protein-coding genes (BUSCO) | 34,406 (97.80%) | 35,810 (98.30%) | 72,410 (99.10%) | 75143 (99.50%) | 74,086 (99.40%) | 72,247 (99.20%) |
| LAI | 24.03 | 21.09 | 21.12 | 21.76 | 21.20 | 21.36 |
| Quality value | 51.57 | 44.97 | 46.15 | 47.78 | 45.24 | 49.38 |
| Repeat content (%) | 74.65 | 77.69 | 76.47 | 76.25 | 76.3 | 76.24 |

of base substitution rates using a rate of $1.64 \times 10^{-8}$ revealed distinct rates of LTR expansion between the At and Bt subgenomes. The *Gypsy* family in the A(t) genome/subgenomes expanded approximately 0.20 million years ago (Ma), before tetraploidization, whereas two expansion peaks occurred at 0.69 Ma and 0.27 Ma in the B(t) genome/subgenomes (Fig. 2a). We observed different insertion times for *Gypsy*-type LTRs in the At and Bt subgenomes. The *Athila*, *Retand*, *CRM* and *Ogre* types experienced two expansions in the B(t) genome/subgenomes, but only the *Reina* type underwent two expansions in the A(t) genome/subgenomes (Supplementary Fig. 4). We categorized *Gypsy* and *Copia* TEs based on their insertion times into four periods: S1 (1–11,000 years ago), S2 (11,001–101,000 years ago), S3 (101,001–301,000 years ago) and S4 (>301,000 years ago). Counts for the A genome were 284, 1,445, 5,323 and 9,553 from S1 to S4; for the B genome, these counts were 467, 1,439, 3,433 and 15,873 (Supplementary Table 10). This indicates that TE insertion events occurred more frequently in the A genome during the S3 period. In the tetraploid subgenomes, counts in the At subgenomes were 343, 1,396, 4,655, and 9,747, whereas those in the Bt subgenomes were 416, 1,392, 3,314 and 15,592, respectively. *Gypsy*-type TEs were primary contributors to TE insertions in the A genome during the S3 phase, with counts of 4,388 in the A genome and 3,861 in the At subgenomes, compared to 2,417 and 2,259 in the B genome and Bt subgenomes, respectively. Comparative analysis of the two main types of *Gypsy* TEs, *Athila* and *Retand*, revealed shorter insertion times in the A(t) genome/subgenomes than in the B(t) genome/subgenomes (Supplementary Fig. 5). Full-length *Athila* phylogenetic trees indicate distinct evolutionary characteristics after the divergence of the A(t) and B(t) genome/subgenomes (Fig. 2b). Notably, several *Athila* and *Retand* TEs were absent from tetraploid subgenomes but present in the diploid genomes (Supplementary Fig. 5).

In the diploid genomes, the total lengths of centromeres were 44.70 Mb and 43.30 Mb; by contrast, the At subgenomes showed an increase in average length of 46.11 Mb, whereas the Bt subgenomes experienced a significant reduction, with an average length of 25.70 Mb (range: 23.18–29.81 Mb) (Supplementary Table 11). This difference was primarily due to a reduction in the number of repetitive units in the Bt subgenomes. Phylogenetic analysis based on centromere monomers revealed that monomers from the A(t) genome/subgenomes clustered together, as did those from the B(t) genome/subgenomes, indicating independent evolution of CentO units (Fig. 2c). Centromeric repetitive elements (CRMs) were interspersed between the At and Bt subgenomes, suggesting synchronous insertion and evolution of CRMs (Fig. 2d). The A(t) and B(t) genome/subgenomes showed different *Retand* element activity, with a later burst insertion observed in the A(t) genome/subgenomes but absent from the B(t) genome/subgenomes (Fig. 2e). The noncollinearity results for *Retand* elements on chromosome 4 indicated that following tetraploid formation, the A genome

and At subgenomes experienced different reshaping in centromere regions (Fig. 2f and Supplementary Fig. 5).

### SVs during peanut evolution and domestication

T2T genomes enable precise identification of SVs and rearrangements through comparison of diploid and tetraploid chromosomes. Using SyRI (v.1.5.3)[26] and SVMU (v.4.0.0beta2)[27], we detected 153,947 insertions, 182,467 deletions, 6,272 copy number variations (CNVs), 2,351 translocations and 644 inversions in four tetraploid peanut accessions compared to diploid genomes V14167 and K30076 (Supplementary Table 12 and Supplementary Fig. 6a,b). Specifically, the At subgenomes exhibited 128,260 insertions, 145,963 deletions, 4,495 CNVs, 2,174 translocations and 520 inversions, whereas the Bt subgenomes showed only 25,687 insertions, 36,504 deletions, 1,777 CNVs, 177 translocations and 124 inversions. Genes located within 2 kb upstream or downstream of the SV breakpoints (termed SV genes) were identified, with an average of 9,682 SV genes per accession (ranging from 9,522 to 10,006); 45.42% of the SV genes were TE-SV genes (Supplementary Table 13 and Supplementary Fig. 7a,b). Further comparisons among tetraploid cultivated accessions revealed a total of 17,216 insertions, 28,554 deletions, 1,692 CNVs, 177 inversions and 215 translocations when comparing S83 to S245, HN873 and HN51 (Supplementary Table 12 and Supplementary Fig. 6c). The At subgenomes averaged 8,370 insertions and 14,712 deletions, whereas the Bt subgenomes had 8,864 insertions and 13,842 deletions. SV gene analysis indicated a total of 4,886, 4,533 and 4,392 SV genes for the S245 versus S83, HN873 versus S83, and HN51 versus S83 comparisons, respectively, with an average of 2,015 classified as TE-SV genes (Supplementary Table 13 and Supplementary Fig. 7c). To validate the accuracy of SV identification, we confirmed 202 large SVs via PCR amplification (Supplementary Table 14 and Supplementary Fig. 8). Gene Ontology (GO) and Kyoto Encyclopedia of Genes and Genomes (KEGG) enrichment analyses of SV genes showed significant associations with pathways involved in flavonoid biosynthesis, brassinosteroid biosynthesis and fatty acid metabolism (Supplementary Tables 15–17). We also identified conserved SVs across all tetraploid subgenomes, indicating rearrangements from or before tetraploidization, particularly in the A(t) genome/subgenomes. By contrast, fewer specific rearrangements were noted in the B(t) genome/subgenomes, with a notable 4.48-fold difference in size between At:A and Bt:B (Supplementary Figs. 9 and 10).

Analysis of the relationship between SVs and domestication revealed unique variations among the tetraploid accessions. Compared to the landrace HN873, S245, HN51 and S83 had significantly more unique SVs and genes. Specifically, S245 contained 7,431 SVs (861 genes), S83 had 6,059 SVs (841 genes) and HN51 had 4,156 SVs (448 genes), whereas HN873 had 2,593 SVs (253 genes). In the GO enrichment analysis, the HN873 genes were significantly enriched

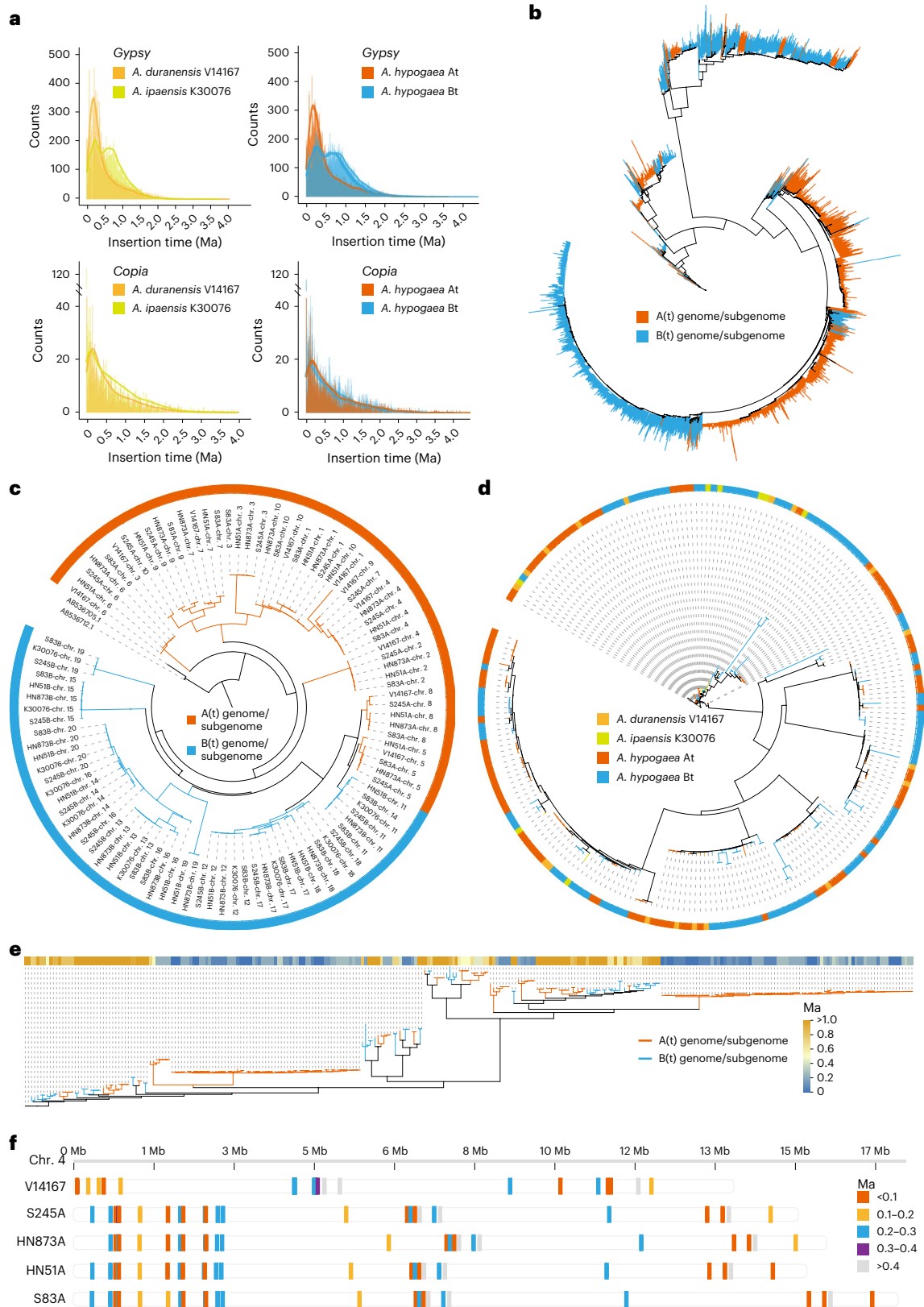

**Fig. 2 | Asymmetric subgenome evolution. a**, Insertion times obtained using the LTR retrotransposons (*Gypsy* and *Copia*) in the *A. hypogaea* At (S245A, HN873A, HN51A and S83A), *A. hypogaea* Bt (S245B, HN873B, HN51B and S83B), *A. duranensis* (V14167) and *A. ipaensis* (K30076) genomes. **b**, Phylogenetic analysis of the expansion of *Athila* LTR retrotransposons across the entire A(t) and B(t) genome/subgenomes. **c**, The phylogenetic relationships between the A(t) and B(t) genome/subgenomes, analyzed using the CentO repeat

unit. **d**, Maximum likelihood phylogenetic tree based on alignments of the full-length CRM within the centromeric regions of the A(t) and B(t) genome/subgenomes. **e**, Phylogenetic relationships and insertion time of full-length *Retand* elements in the centromeric regions of the A(t) and B(t) genome/subgenomes. **f**, Synteny analysis and timing of insertion of full-length *Retand* elements in the At subgenomes and the diploid genome (V14167) in the centromeric region on chr. 4.

in terms including calcium ion transport, metal ion transport and and monoatomic cation transport, whereas HN51 was linked to tissue development terms such as embryonic morphogenesis, flower development, pollen tube growth, post-embryonic development, shoot system development, vascular transport and phloem transport (Supplementary Table 18). Unique SV genes in S245 and S83 were enriched in fatty acid synthesis and metabolism, with S245 also enriched in defense responses and nutrient level regulation, whereas S83 was related to root development and endoderm formation. These findings suggest that SVs have a crucial role in shaping the domestication of different peanut varieties through artificial selection.

A 13.23-Mb insertion on chr. 14 was found in the K30076, S83, HN51 and HN873 accessions (Fig. 3a–e), with next-generation sequencing showing the insertion to be present in all 161 var. *hirsuta* but only 33 of 161 var. *hypogaea* accessions (Supplementary Fig. 11). This insertion could contribute to phenotypic differences between var. *hirsuta* and var. *hypogaea*. KEGG analysis of the 109 genes within the insertion regions highlighted enrichments in lipid metabolism, lignin biosynthesis and responses to environmental stressors (Fig. 3f,h and Supplementary Fig. 12). Genes of chr. 14.1804 homologous to *OsLAC* (Fig. 3g,i), which plays an important part in regulation of plant architecture and seed development in rice[28], were expressed in peanut lateral branch, axillary bud and hypocotyl (Supplementary Fig. 12). Another gene, chr. 14.1829, which is homologous to *WOX3A/B*[29,30], was specifically expressed in the embryo, axillary buds and stem apices (Fig. 3g,i and Supplementary Fig. 12), suggesting that the insertion may have a role in the unique phenotype of var. *hirsuta*.

## Population structure

A total of 521 globally collected peanut accessions were sequenced, yielding an average depth of 16.76× and 96.71% genome coverage (Supplementary Table 19). Using the S245 genome as a reference, we identified 101,334,387 high-quality single-nucleotide polymorphisms (SNPs), averaging 39 SNPs per kilobase (Supplementary Table 20). Phylogenetic and population structure analyses classified the accessions into five genetic groups: wild (G1), var. *fastigiata* + admixture (G2), var. *vulgaris* (G3), var. *hypogaea* (G4) and var. *hypogaea* + var. *hirsuta* (G5) (Fig. 4a and Supplementary Figs. 13 and 14a). Principal component analysis (PCA) confirmed the five distinct clusters corresponding to lineages G1–G5 (Fig. 4b). The G5 population showed significant differentiation from G2 and G3 but not from G4. Genetic diversity analysis indicated that the most pronounced differences were between G2 and other groups, whereas G4 and G5 exhibited minimal differences (Fig. 4c). The linkage disequilibrium (LD) decay rate was lowest in G1 and highest in G2 (Fig. 4d). Divergence times estimates indicated that cultivated peanuts diverged into var. *fastigiata* (G2+G3) and var. *hypogaea* (G4+G5) around 9,400 years ago, with further differentiation occurring among groups at approximately 7,200 and 8,900 years ago, respectively (Supplementary Fig. 14b).

## Selected regions and introgressed tracts

Genomic regions under selection were analyzed by calculating population fixation statistic ($F_{ST}$) and cross-population composite likelihood ratio (XP-CLR) values between groups (Supplementary Tables 21 and 22). The top 1% of $F_{ST}$ and XP-CLR intervals were retained to identify selected genes within the G2–G3, G2–G4, G2–G5, G3–G4, G3–G5 and G4–G5 groups. TE insertions were categorized into five time periods for assessment of their impact on these selected sweep regions; this showed that recent TE insertions had smaller effects compared to earlier ones (Supplementary Fig. 15). After filtering of TE intervals, the average length of selective sweep regions in the top 1% of $F_{ST}$ was 69.39 Mb, containing an average of 2,463 genes (ranging from 697 to 3,838) (Supplementary Table 21). Notably, an average 4.98 Mb of differentially selected regions were identified in the Bt subgenomes, compared to only 2.47 Mb in the At subgenomes, indicating asymmetric evolution.

Four-taxon fd statistics detected introgressed regions, with significant gene flow primarily observed in the At subgenome, especially in the G2–G3 and G4–G5 groups (Fig. 4e–g, Supplementary Fig. 16 and Supplementary Tables 23 and 24). Compared to diploids, 22.48%, 51.97% and 49.19% of SNPs were retained in the top 1% of $F_{ST}$, XP-CLR and introgression regions in tetraploids, respectively (Supplementary Figs. 17 and 18). GO and KEGG analyses identified selected genes related to fatty acid synthesis, flavonoid biosynthesis and nutrient metabolism (Supplementary Fig. 19).

## Identification of peanut haplotypes linked to higher oil content

Genome-wide association study (GWAS) analysis identified significant signals on chromosome 8 associated with peanut oil content (Fig. 5a–c), pinpointing 151 genes in the 200-kb regions. Among these, we identified a candidate gene, chr. 8.2620 (named *AhWRI1*), located approximately 54,934 bp from the peak at chr. 8:54293412 ($P = 3.02 \times 10^{-8}$). Gene structure analysis revealed a base change (A to C) in the promoter region and a nonsynonymous SNP in the first exon (T to G) that altered an amino acid from arginine to methionine (Fig. 5d–f). Expression analysis of *AhWRI1* indicated predominant expression during seed development (Supplementary Fig. 20). Based on 153 accessions with RNA sequencing (RNA-seq) data, the Hap1 type had an average fragments per kilobase million (FPKM) value of 16.30 and an oil content of 48.41%, whereas the Hap2 type had an average FPKM value of 24.81 and an oil content of 54.10% (Supplementary Table 25). Quantitative PCR with reverse transcription (RT–qPCR) confirmed higher *AhWRI1* expression in Hap2-type accessions (Supplementary Fig. 21). The correlation between promoter haplotype and expression level suggested causal variation in the promoter region. Subcellular localization studies confirmed that AhWRI1 is a transcription factor (Fig. 5g). Transgenic lines overexpressing *AhWRI1* in rapeseed and soybean showed lipid droplet accumulation in the leaf mesophyll cells, as detected by confocal imaging and Nile red staining (green) (Fig. 5h,i and Supplementary Fig. 22a). We assessed total triacylglycerol content in seeds and found that *AhWRI1* overexpression significantly increased fatty acid levels (Fig. 5j and Supplementary Fig. 22b). We also noted that four transcriptional regulators, LEAFY COTYLEDON1 (LEC1), FUSCA3 (FUS3) and ABSCISIC ACID INSENSITIVE3 (ABI3), were crucial for seed development[31–35] and oil production[36–38]. We found that *AhLEC1*, *AhFUS3*, *AhABI3* and *AhWRI1* had higher expression in high-oil varieties compared to low-oil ones (Supplementary Fig. 21). Yeast one-hybrid and dual-luciferase assays showed that *AhWRI1* regulated downstream plant fatty acid biosynthesis pathway protein genes *AhACP1* and *AhKAS1*, whereas *AhFUS3*, *AhABI3* and *AhLEC1* bound to the *AhWRI1* promoter (Fig. 5k,l). The E-box motif, identified in the phase promoter of beans with respect to its strict expression during embryogenesis[39], was also recognized as an induced binding motif in *cis*-motifs associated with ABI3 binding peaks in *Arabidopsis*[36]. An A-to-C mutation was discovered in the 2,040-bp promoter region of *AhWRI1*, located within an E-box, with a conserved RY motif on the complementary strand recognized by seed-specific transcription factors (Fig. 5m). In yeast one-hybrid assays, cloning of the 52-bp sequence containing the RY and E-box motifs revealed that the C-to-A mutation eliminated interaction with FUS3 (Fig. 5n). In addition, dual-luciferase assays showed that the promoter activity of the C variant was significantly higher than that of the A variant, regardless of combination with SK ($P = 4.14 \times 10^{-5}$, $n = 6$) or FUS-SK ($P = 1.04 \times 10^{-6}$, $n = 6$) (Fig. 5o). This mutation affected binding with *AhFUS3*, leading to variations in transcriptional activity and expression of the *AhWRI1* gene that resulted in differences in average oil content among varieties.

### *AhGSA1* gene for increased peanut seed size and weight

Seed size and weight are critical traits in peanut breeding, yet understanding of the natural causal variation underlying these traits remains limited despite numerous identified QTLs. Through a GWAS, we

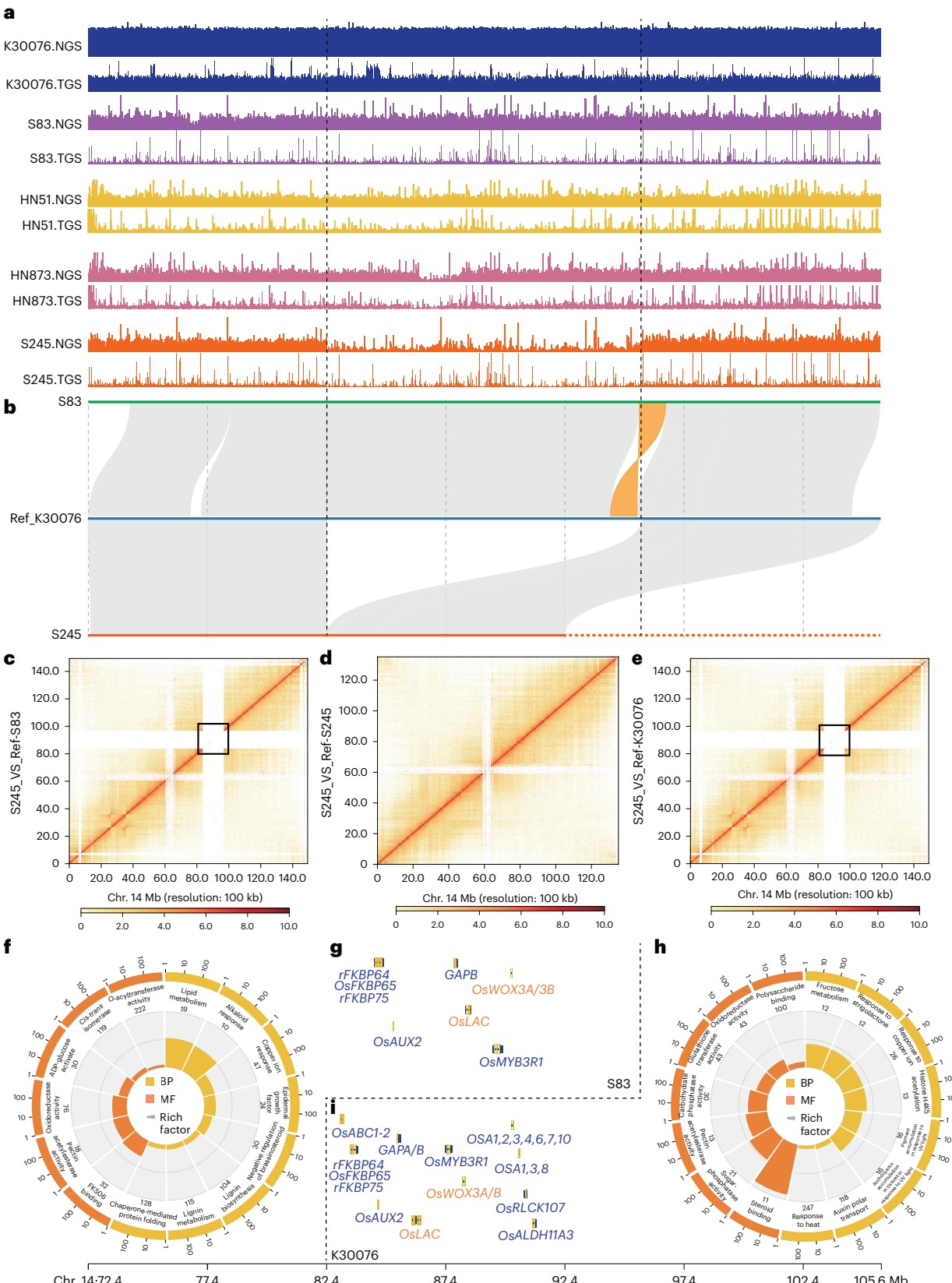

**Fig. 3 | SVs among tetraploid peanuts. a**, Coverage of SVs on chr. 14 of the K30076, S83, HN51, HN873 and S245 genomes using Illumina (next-generation sequencing) and HiFi (third-generation sequencing) reads. **b**, A large SV (deletion) was detected in S245 during collinearity analysis of S83, K30076 and S245. **c**–**e**, Hi-C heatmaps constructed using reads from the S245 accession and the reference genomes for S83 (**c**), S245 (**d**) and K30076 (**e**). **f**, GO enrichment analysis of genes located in the SV regions of the S83. **g**, Functional homologous rice genes of the SV genes in S83. **h**, GO enrichment analysis of genes located in the SV regions of K30076. **i**, Functional homologous rice genes of the SV genes in K30076. NGS, next-generation sequencing; TGS, third-generation sequencing; UV, ultraviolet.

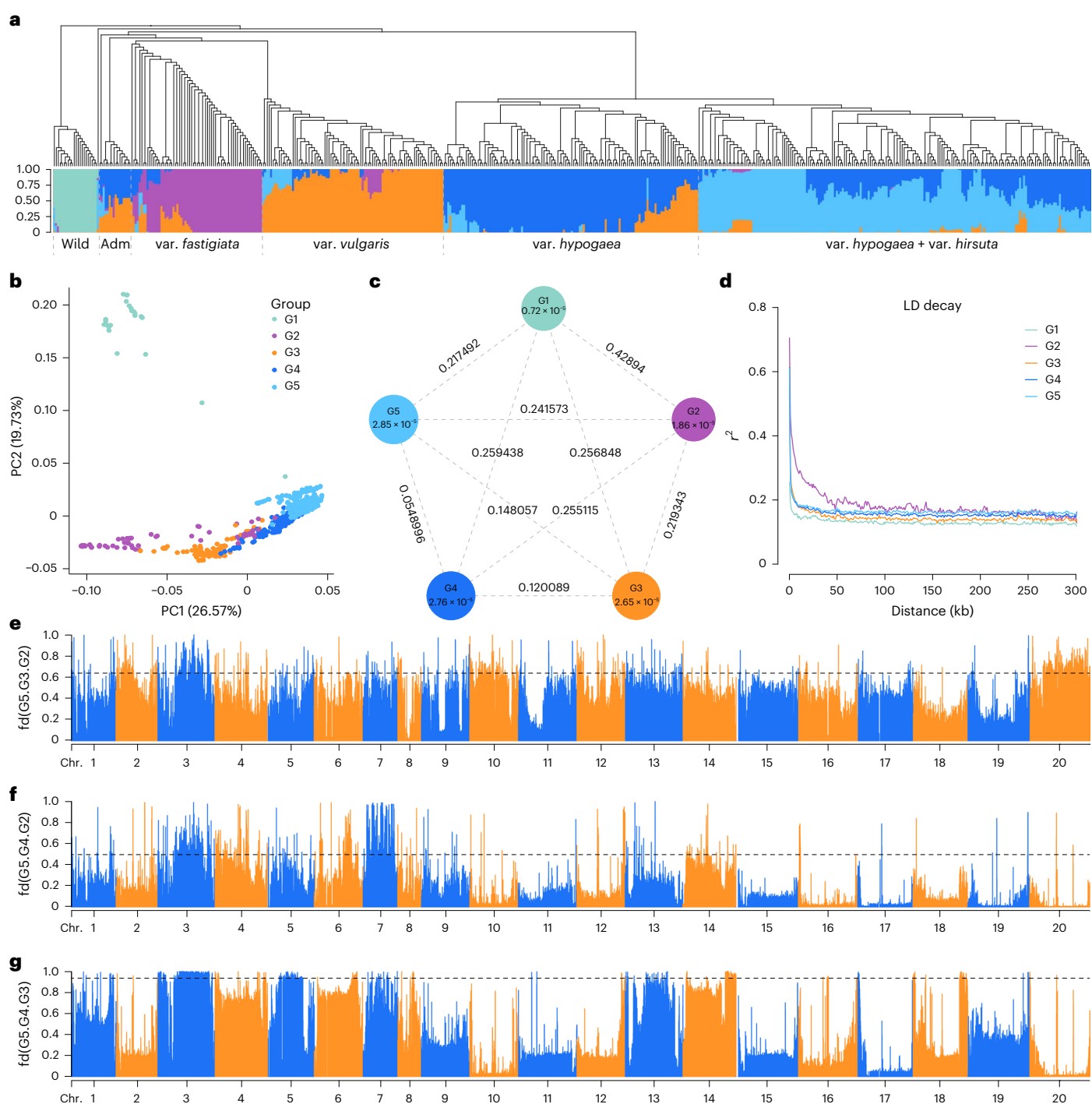

**Fig. 4 | Population structure and introgression analyses. a**, Phylogenetic tree and population structure analysis of 521 peanut accessions. **b**, PCA plot of the first two principal components (PC1 and PC2) of the five groups. **c**, Population differentiation ($F_{ST}$) and genetic diversity ($\pi$) across the five groups. **d**, LD decay estimation for the five groups. **e–g**, Introgression analysis (fd) among different groups: G2 to G3 (top 1% threshold line: 0.64 (**e**)), G2 to G4 (top 1% threshold line: 0.49 (**f**)) and G3 to G4 (top 1% threshold line: 0.94 (**g**)) on different chromosomes, with G5 as the outgroup. Adm, admixture.

identified 62 candidate SNPs on chromosome 16, spanning 130 genes (Fig. 6a–d). These genes included chr.16.3093 (named *AhGSA1*), annotated as glycosyltransferase, overexpression of which significantly increased grain size by modulating cell proliferation and expansion in rice[40]. Notably, an insertion–deletion within the promoter region (660 bp from the transcription start site) of *AhGSA1* was identified (Fig. 6e). RT–qPCR analysis of 30 small-seeded and 30 large-seeded accessions further confirmed a significant difference in the expression levels of Hap1 and Hap2 (Fig. 6f). Forty-two accessions classified as Hap1

(ATT type) had an average thousand-grain weight of 490.82 g, whereas 293 accessions of the Hap2 (AT type) exhibited a significantly higher average thousand-grain weight of 846.34 g ($P = 1.00 \times 10^{-20}$) (Fig. 6g). In addition, expression analysis showed that *AhGSA1* had high expression levels in peanut pods at all developmental stages but was not expressed in the seed coat (Supplementary Fig. 23). The correlation between haplotypes in the promoter region and expression levels suggested that *AhGSA1* contributes to variation in seed size and weight. Furthermore, Hap1 was predominantly found in the G2 (var. *fastigiata*) population,

indicating differential selection among cultivated groups (Fig. 6h). We cloned the 1,000-bp promoter sequence harboring the ATT or AT mutation and demonstrated its ability to activate GUS reporter gene expression. The GUS expression driven by the AT promoter was stronger than that driven by the ATT promoter (Fig. 6i). Subcellular localization studies showed that AhGSA1 was localized to the cytoplasmic membrane (Fig. 6j). Dual-luciferase analysis confirmed that AT promoter activity was significantly greater than that of ATT ($P = 1.02 \times 10^{-4}$, $n = 3$) (Fig. 6k), highlighting the importance of the insertion–deletion in the promoter region for *AhGSA1* transcription.

### Gene networks for lipids and anthocyanins in seed development

Peanut seeds have a high oil content, accounting for approximately 50–80% of their weight. We performed integrated transcriptomic and metabolomic analyses of S83 and S245 at five developmental stages (Fig. 7a,b and Supplementary Tables 26–30). A total of 497 lipid metabolites were identified, including 300 glycerolipids, 153 glycerophospholipids, 24 sphingolipids and 18 fatty acids (Fig. 7c and Supplementary Table 27). Clustering based on lipid content revealed that S83 consistently had higher lipid levels than S245, especially during later developmental stages (Fig. 7d). Lipid accumulation, which is critical for seed development, is regulated by complex metabolic pathways (Fig. 7e). Distinct accumulation patterns for lipid metabolism were noted in both varieties, peaking at the end of stage S5 (Fig. 7f). To identify key genes related to lipid metabolism, we conducted weighted gene coexpression network analysis (WGCNA); this identified 11 coexpressed gene modules (Supplementary Fig. 24a,b). The module most correlated with lipid metabolite content contained several key genes, including those encoding fatty acid desaturase and ketoacyl-ACP synthase, which are crucial for fatty acid biosynthesis (Supplementary Table 28). Phylogenetic analysis of these enzymes revealed evolutionary relationships that could explain their roles in lipid biosynthesis (Supplementary Fig. 25a,b). Their expression patterns matched metabolite level trends, peaking during development (Fig. 7g).

We also investigated anthocyanidin metabolism and identified 78 metabolites across eight classes (Fig. 7h and Supplementary Table 29). Clustering showed that S83 had higher concentrations of petunidin, malvidin, flavonoids and delphinidin, whereas S245 had more procyanidin, peonidin, pelargonidin and cyanidin (Fig. 7i). These metabolites could influence differences in seed coat color between S83 and S245 (Fig. 7j,k). WGCNA identified key genes with expression levels that were correlated with the abundances of differentially accumulated metabolites during seed development (Supplementary Fig. 24c,d), in particular, genes encoding the MYB and bHLH transcription factors, which are known to regulate anthocyanin biosynthesis[19,41,42] (Supplementary Table 30). Phylogenetic analysis of these transcription factors, including those previously reported, provided insights into their evolutionary relationships and potential functional divergence (Supplementary Fig. 25c,d). The expression patterns of these key transcription factors were consistent with the accumulation trends of anthocyanin metabolites, implying roles in the regulation of pigment biosynthesis during seed development (Fig. 7l).

In summary, our analyses identified important genes involved in lipid and anthocyanin biosynthesis during peanut seed development. These genes may have central roles in determining oil content and seed coat color and could thus represent targets for future studies and crop improvement.

## Discussion

T2T genome assemblies represent an important advance in biological studies and crop breeding. However, the peanut genome poses challenges owing to characteristics including allopolyploidy, high repetition rates and large size. Here we present six high-quality T2T genomes of representative peanut accessions, following the path established for the human genome[43] and model plants such as *Arabidopsis*[44,45], rice[46], maize[47] and soybean[48]. The asymmetric insertion of TEs is crucial for genomic shock. Our research indicates that tetraploids show a higher frequency of recent TE insertions in the At subgenomes compared to the Bt subgenomes, along with TE deletions. Significant TE insertions in the At subgenomes occurred around 101,001–300,000 years ago, probably facilitating environmental adaptation. TE-associated SVs suggest that many originated from TE mobilization, affecting agronomic traits such as pod size and oil content. Similar patterns have been noted in cotton[49] and soybean[50], in which asymmetric evolution of subgenomes is correlated with fiber characteristics, seed oil content, weight and yield.

Optimal centromere length enhances chromosome alignment and stability, influencing effective recombination in cultivated varieties. Phylogenetic analysis showed that the A genome clusters at the base of the tree, indicating that a diploid B species may have been derived from a diploid A species. In addition, a recent expansion event in the centromeric regions of the At subgenome was evident. These T2T genomes facilitate deeper understanding of centromeric evolution in peanuts.

Understanding the relationship between SVs and domestication is vital for determining the genetic basis of adaptive traits. We identified a 13.23-Mb variety-specific SV on chromosome 14, which was present in all G5 group accessions but only in 23% of G4 and 15% of G3 groups, and absent from the G2 group. This SV included multiple genes related to plant architecture, indicating a role in differentiation of G4 and G5 groups. Ongoing selection of key genes has contributed to the wide cultivation and utilization of peanuts. Selected sweeps and introgression analysis showed that the At subgenomes exhibited low differentiation and high gene flow, whereas the Bt subgenomes had high differentiation and low gene flow. This pattern suggests the occurrence of multiple hybridization and introgression events in cultivated peanut varieties, particularly affecting the Bt subgenomes. Haplotype variations

**Fig. 5 | Causal gene and haplotype controlling oil content. a**, Genome-wide screening of selective sweep regions using $F_{ST}$ analysis (F-statistics). The top 1% threshold line is at 0.57. **b**, Significant signals from a GWAS of oil-related traits on chr. 8. Horizontal lines represent the significance threshold ($P = 6.26 \times 10^{-7}$, Bonferroni correction). **c**, LD block of a candidate peak with 200-kb upstream and downstream regions. **d**, Structure of the *AhWRI1* gene and two variants in the promoter and the first exon. **e**, Significant difference in oil content between the Hap1 (range: 44.56–52.00%; $n = 60$) and Hap2 (range: 42.38–57.23%; $n = 436$) accessions. Center line, median; box lower and upper edges, 25% and 75% quartiles, respectively; whiskers, 1.5× interquartile range; colored dots, outliers. **f**, The distribution of Hap1 and Hap2 across different groups. **g**, Subcellular localization of AhWRI1. **h**, Accumulation of lipid droplets in rapeseed leaf mesophyll cells overexpressing *AhWRI1*. **i**, Confocal images of Nile red-stained lipid droplets (LDs; green) in rapeseed leaves with the trans-*AhWRI1* gene. **j**, Total triacylglycerol content in rapeseed leaves expressing AhWRI1. Quantitative data are mean ± s.e.m. $n = 9$ biologically independent samples. **k**, Yeast one-hybrid assays showing binding of AhWRI1 to the promoters of *AhACP1* and *AhKAS1* and binding of *AhFUS3*, *AhABI3* and *AhLEC1* to the *AhWRI1* promoter. **l**, Dual-luciferase assays in *N. benthamiana* leaves showing that *AhWRI1* activates the *AhACP1* and *AhKAS1* promoters, and that *AhFUS3*, *AhABI3* and *AhLEC1* activate the *AhWRI1* promoter; an empty reporter (no promoter) served as a negative control. Quantitative data are mean ± s.e.m. $n = 6$ biologically independent samples. **m**, Mutation site in the *AhWRI1* promoter; the E-box motif is highlighted in blue and the RY motif in pink. **n**, Yeast one-hybrid assay showing that the C-to-A substitution in the *AhWRI1* promoter abolishes interaction with *AhFUS3*. **o**, Dual-luciferase assay comparing activities of the *AhWRI1* promoters carrying the A or C allele. Quantitative data are mean ± s.e.m. $n = 6$ biologically independent samples. $P$ values were calculated by two-tailed Student's *t*-tests (**e**, **j**, **i** and **o**). TAG, triacylglycerol; TSS, transcription start site; UTR, untranslated region; WT, wild type.

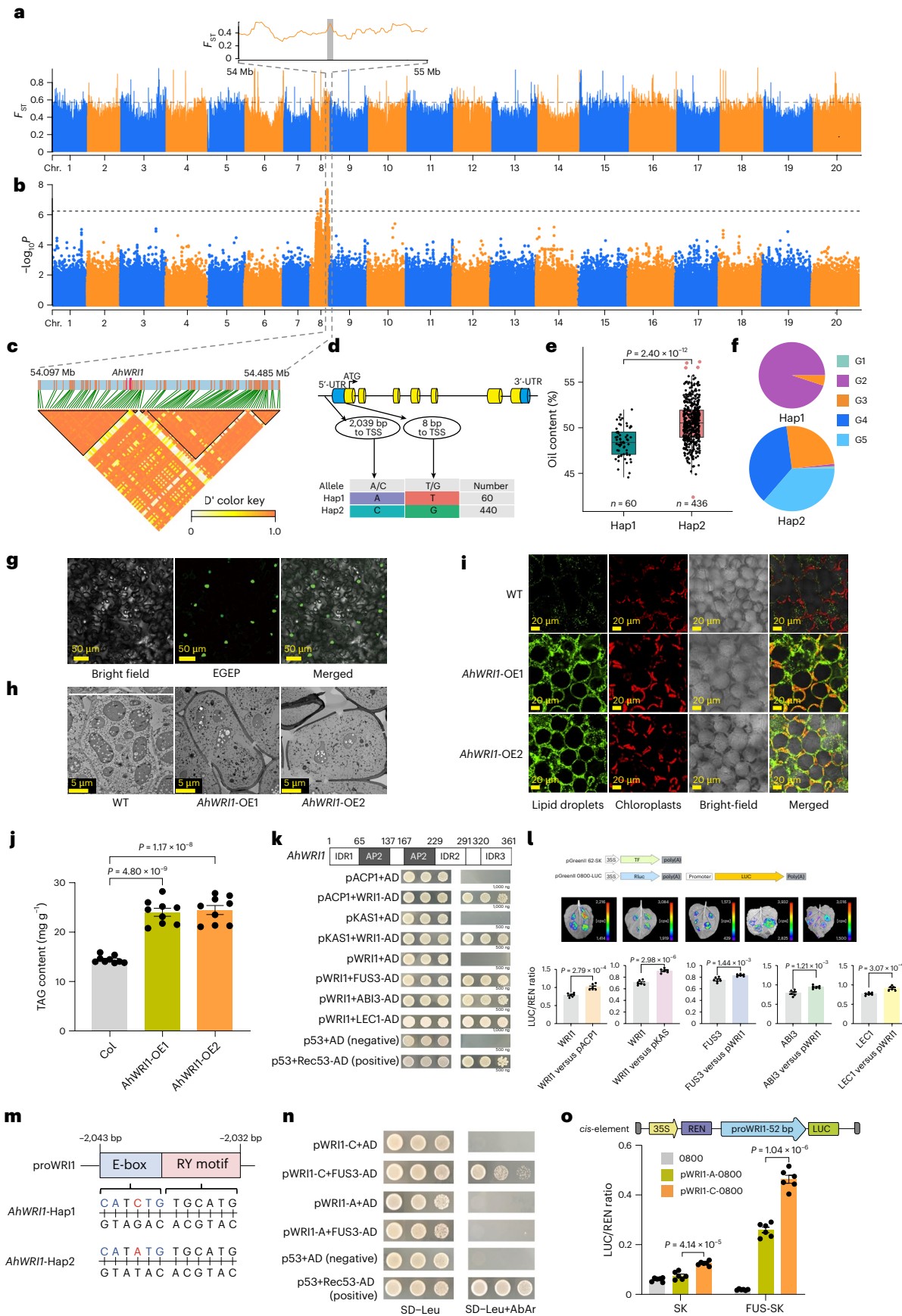

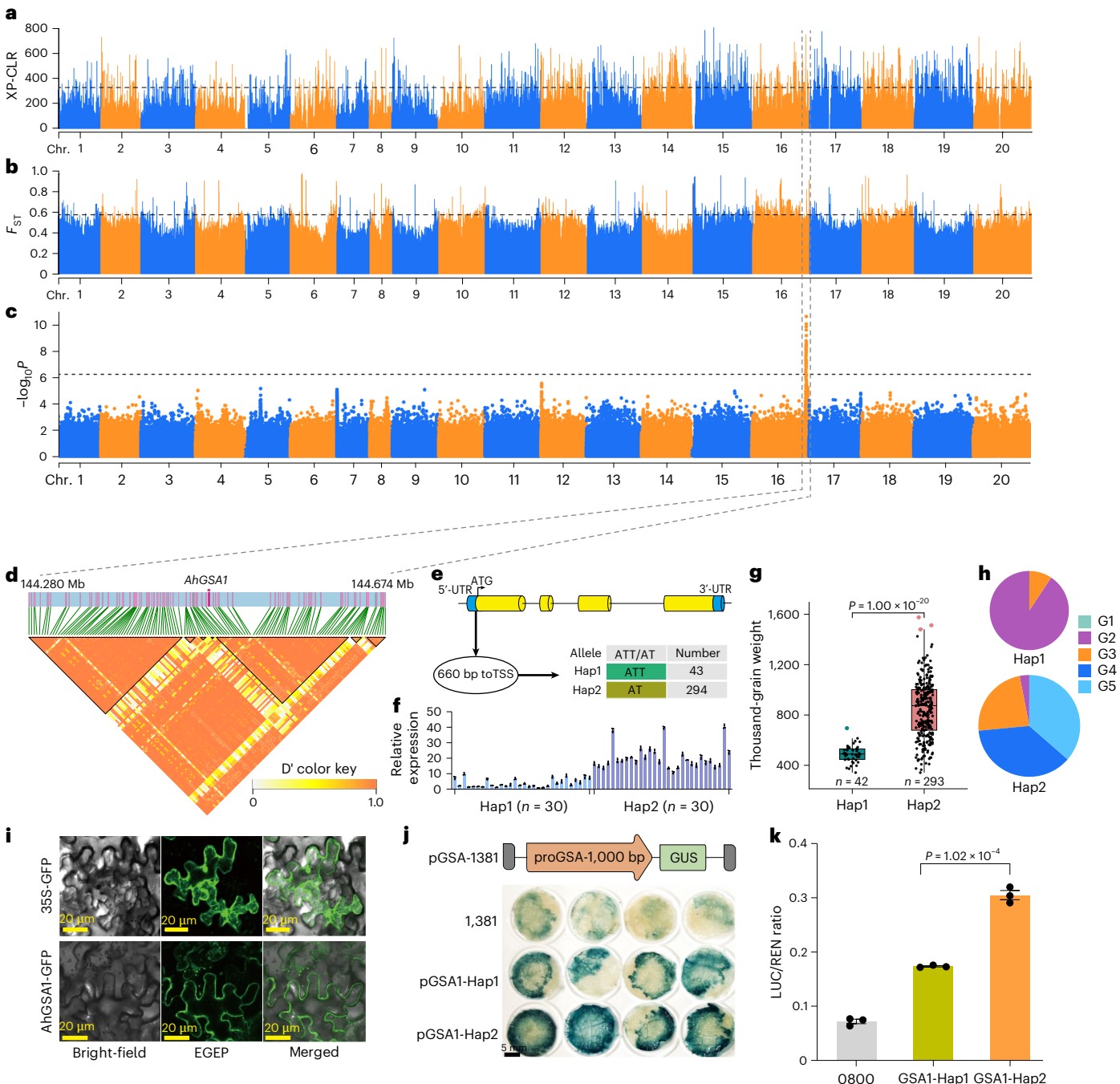

**Fig. 6 | Candidate gene associated with peanut seed size and weight.**
**a**,**b**, Genome-wide scans for selective sweeps using XP-CLR test (threshold line: 326.54) (**a**) and $F_{ST}$ (threshold line: 0.58; F-statistics) (**b**). **c**, Significant signals from GWAS of oil-related traits on chr. 16. Horizontal lines represent the significance threshold ($P = 6.26 \times 10^{-7}$, Bonferroni correction). **d**, LD block around the candidate peak within a 200-kb interval. **e**, Gene structure and promoter variant of *AhGSA1*. **f**, RT–qPCR analysis of *AhGSA1* expression in 30 Hap1 and 30 Hap2 accessions. Quantitative data are mean ± s.e.m. $n = 3$ biologically independent samples. **g**, Box plot showing significant differences in

thousand-grain weight between Hap1 (range: 342–693; $n = 42$) and Hap2 (range: 342–1,563.64; $n = 293$) accessions. Center line, median; box lower and upper edges, 25% and 75% quartiles, respectively; whiskers, 1.5× interquartile range; colored dots, outliers. **h**, Distribution of Hap1 and Hap2 across different groups. **i**, Subcellular localization of AhGSA1. **j**, Histochemical GUS staining driven by the Hap1 and Hap2 *AhGSA1* promoters. **k**, Dual-luciferase assay comparing Hap1 and Hap2 promoter activities. Quantitative data are mean ± s.e.m. $n = 3$ biologically independent samples. *P* values were calculated by two-tailed Student's t-tests (**g** and **k**).

reflect domestication pathways. The var. *fastigiata* (G2+G3) has significantly smaller seeds than var. *hypogaea* (G4+G5). The small-seed haplotype (Hap1) of *AhGSA1* was found only in var. *fastigiata*, whereas the large-seed haplotype (Hap2) was found mainly in var. *hypogaea*. Notably, although var. *fastigiata* traditionally has higher oil content, the high-oil haplotype (Hap2) of *AhWRI1* was predominantly found

in var. *hypogaea*, suggesting that modern breeding prioritizes both yield and oil content.

We also conducted transcriptomic and metabolomic analyses of two peanut varieties, S83 and S245, across five developmental stages to investigate seed development. A total of 497 lipid metabolites were detected, with S83 consistently exhibiting higher lipid contents than

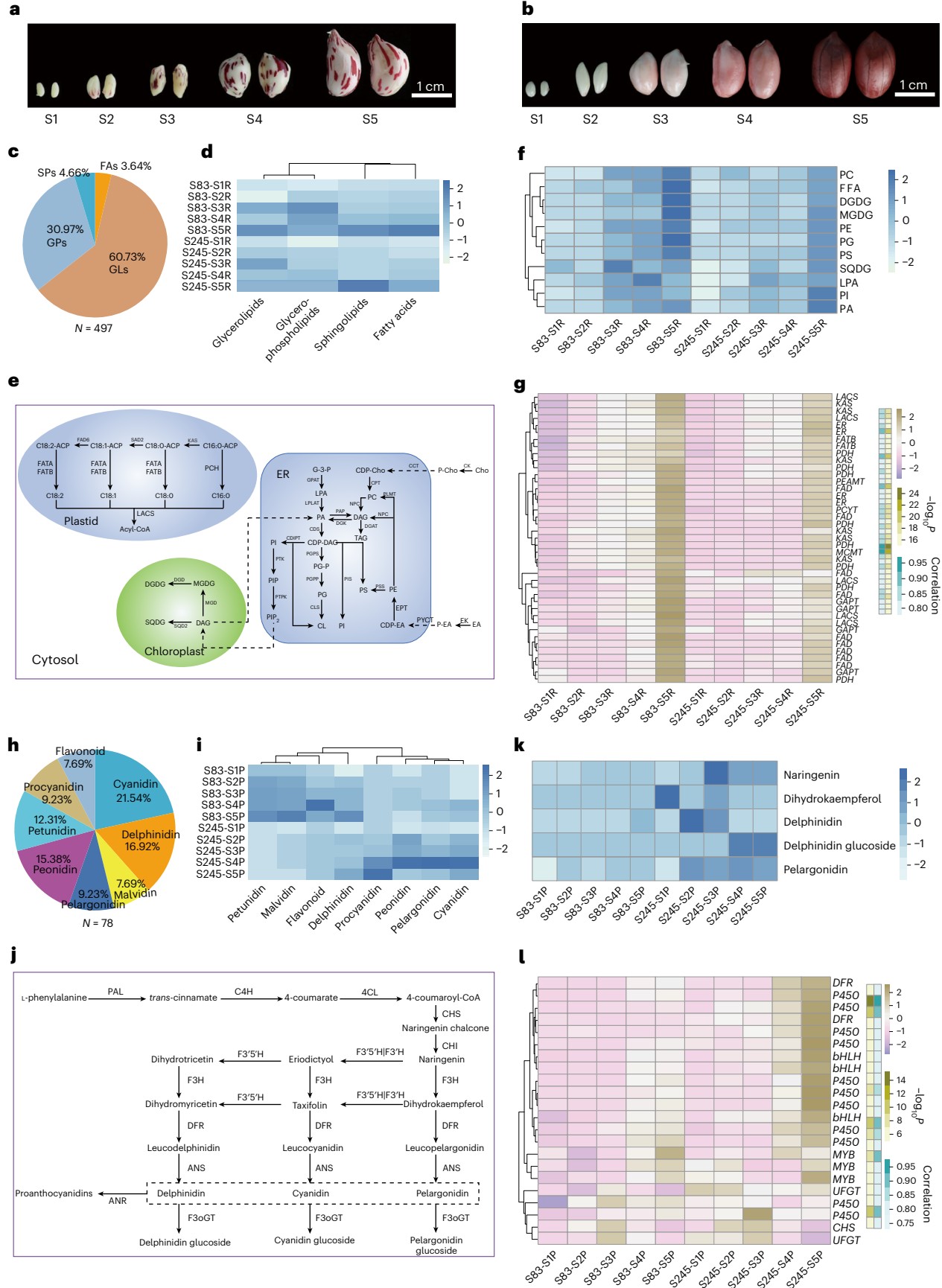

**Fig. 7 | Transcriptomic and metabolomic analysis during seed development.**
**a,b**, Seed diversity between two peanut subpopulations across five developmental stages for S83 (**a**) and S245 (**b**). **c**, Lipid metabolites (497) categorized into glycerolipids, glycerophospholipids, sphingolipids and fatty acids. **d**, Clustered heatmap of the levels of four classes of lipid in S83 and S245 across five developmental stages. **e**, Schematic of lipid metabolic pathways, with detected metabolites marked in bold black. **f**, Clustered heatmap displaying the contents of metabolites involved in the lipid metabolism pathway for S83 and S245. **g**, Heatmap of expression levels of candidate lipid metabolism genes; the correlation between gene expression and metabolite contents is indicated in teal, and significance is denoted by the olive gradient. **h**, Anthocyanidin contents of 78 metabolites with anthocyanins categorized into eight classes. **i**, Clustered heatmap of eight classes of anthocyanidins in S83 and S245 across five developmental stages. **j**, Schematic of anthocyanidin biosynthesis pathways, with detected metabolites marked in bold black. **k**, Clustered heatmap

displaying the contents of candidate metabolites contributing to the difference in color between S83 and S245. **l**, Heatmap of the expression levels of candidate anthocyanidin biosynthesis genes; the correlation between gene expression and metabolite contents is indicated in teal, and significance is denoted by the olive gradient. CD, cardiolipin; CDP-DAG, cytidine diphosphate diacylglycerol; CDP-EA, cytidine diphosphoethanolamine; Cho, choline; DAG, diacylglycerol; DGDG, digalactosyldiacylglycerol; EA, ethanolamine; ER, endoplasmic reticulum; FAs, fatty acids; FFA, free fatty acids; G3P, glycerol-3-phosphate; GLs, glycerolipids; GPs, glycerophospholipids; LPA, lysophosphatidic acid; MGDG, monogalactosyldiacylglycerol; PA, phosphatidic acid; PC, phosphatidylcholine; P-Cho, phosphocholine; PE, phosphatidylethanolamine; P-EA, phosphoethanolamine; PG, phosphatidylglycerol; PG-P, phosphatidylglycerol phosphate; PI, phosphatidylinositol; PIP2, phosphatidylinositol 4,5-bisphosphate; PS, phosphatidylserine; SPs, sphingolipids; SQDG, sulfoquinovosyldiacylglycerol; TAG, triacylglycerol.

S245. Using WGCNA, we identified several enzymes linked to lipid metabolism, which were expressed at higher levels in S83. Furthermore, distinct anthocyanin profiles corresponded to the seed coat colors of S83 and S245. Differences in expression of the *AhRt2* gene[19,51], which encodes a MYB transcription factor associated with testa color, suggested a role in control of the color variation between these varieties. Our findings highlight unique metabolic pathways and synthesis mechanisms that contribute to peanut seed development and coloration, providing insights into the molecular processes driving these traits.

## Online content

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

[1]Peking University Institute of Advanced Agricultural Sciences, Shandong Laboratory of Advanced Agricultural Sciences in Weifang, Weifang, China. [2]Peking-Tsinghua Center for Life Sciences, School of Life Sciences and School of Advanced Agricultural Sciences, Peking University, Beijing, China. [3]WA State Agricultural Biotechnology Centre, Centre for Crop and Food Innovation, Food Futures Institute, Murdoch University, Murdoch, Western Australia, Australia. [4]Guangxi Academy of Agricultural Sciences, Nanning, China. [5]Institute of Crop Germplasm Resources (Institute of Biotechnology), Shandong Academy of Agricultural Sciences, Jinan, China. [6]Innovation Center of Yangtze River Delta, College of Agriculture and Biotechnology, Zhejiang University, Hangzhou, China. [7]These authors contributed equally: Jianxin Bian, Yilin Zhang, Shuai Ding, Haosong Guo. ✉e-mail: rajeev.varshney@murdoch.edu.au; hehang@pku.edu.cn; xiaoqin.liu@pku-iaas.edu.cn

## Methods

### Plant materials

Detailed information about the 521 peanut accessions, including type (wild, landrace and cultivated), ploidy (diploid and tetraploid), subspecies (var. *hypogaea* and var. *fastigiata*), and variety (var. *fastigiata*, var. *vulgaris*, var. *hypogaea*, var. *hirsute*, var. *peruviana* and var. *aequatoriana*) used in this project is provided in Supplementary Table 19. Briefly, the two diploid progenitors, V14167 and K30076, were *A. duranensis* (AA) *and A. ipaensis* (BB) genotypes collected from the Guangxi Provincial Academy of Agricultural Sciences. Laiyang Silihong (S245, subspecies var. *fastigiata*), a representative peanut variety known for its high seed count (four seeds), was collected from Laiyang, Shandong Province, North China. Chedouzi (HN873, subspecies var. *vulgaris*), a landrace accession, was collected from Nanchong, Sichuan Province, Southwest China. Huayu23 (HN51, subspecies var. *hypogaea*) is a widely cultivated oil-type peanut variety developed through hybridization and selection by the Shandong Peanut Research Institute. Yunnan Rainbow Peanut (S83, subspecies var. *hirsute*) was named for the color variations in its seed coat. The research presented here complied with the Nagoya protocol.

### Genomic DNA extraction, library construction and sequencing

Twenty plants of each accession were grown at the Xiashan Experimental Base of the Peking University Institute of Advanced Agricultural Sciences in Weifang, Shandong. Young leaf samples were collected 30 days after sowing, rapidly frozen in liquid nitrogen, and stored at −80 °C for genomic DNA extraction using the cetyltrimethylammonium bromide protocol. For PCR-free library preparation, 0.2 µg of DNA per accession was used. Libraries underwent phosphorylation and cyclization, generating DNA nanoballs that were sequenced on the DNBSEQ-T7 platform at Novogene. High-molecular-weight DNA for genome sequencing was isolated from *A. duranensis*, *A. ipaensis* and four *A. hypogaea* cultivars, with paired-end libraries prepared for DNBSEQ. For PacBio HiFi sequencing, genomic DNA was purified and used to construct HiFi SMRTbell libraries, whereas ONT ultralong sequencing libraries were prepared using a SQK-LSK110 ligation kit.

### Hi-C library construction and sequencing

The Hi-C library was constructed following a published method[52,53]. In brief, library construction involved the following steps: cross-linking of cells with formaldehyde, digestion of DNA with a suitable 4-cutter restriction enzyme (DpnII), filling of ends and labeling with biotin, ligation of the resulting blunt-end fragments, and purification and random shearing of DNA into 300- to 500-bp fragments. Following quality control and assessment using Qubit 2.0 and an Agilent 2100 instrument (Agilent Technologies) and qPCR, 150-bp paired-end sequencing of the Hi-C library was conducted on an Illumina NovaSeq 6000 platform by Berry Genomics.

### RNA library construction and sequencing

Twenty replicate plants of each of the six accessions were grown in a climate-controlled chamber at 25 °C with a 16-h light/8-h dark cycle. Tissues at different developmental stages (Supplementary Table 6) were sampled and frozen in liquid nitrogen. Total RNA was isolated from different tissues of six accessions using TRIzol. RNA integrity and quality were assessed with a NanoDrop 2000 and an Agilent 2100 Bioanalyzer. For Illumina sequencing, libraries were generated using a NEBNext Ultra RNA Library Prep Kit and sequenced on an Illumina platform, producing 150-bp paired-end reads. For Iso-Seq, high-quality RNA (2 µg) from all developmental stages was used to construct the PacBio SMRTbell library, followed by purification and size selection before sequencing on the Sequel II system.

### Genome assembly, gap closure and polishing

A flowchart for the genome assembly is shown in Supplementary Fig. 26. Raw sequencing reads were processed with fastp (v.0.22.0)[54] to remove adapters and low-quality sequences. Hi-C data were classified into valid interaction pairs using HiC-Pro (v.2.11.1)[55]. Genome sizes for V14167, K30076, S245, HN873, HN51 and S83 were estimated based on 17-mers derived from Illumina data. Hifiasm (v.0.19.5)[22] was used to generate preliminary assemblies from long HiFi reads and valid Hi-C reads, and valid Hi-C data were used to anchor large contigs (>1 Mb) onto chromosomal scaffolds with the 3D-DNA pipeline[56]. We validated the anchored scaffolds with Tifrunner[2] genome as reference using NUCmer[57] and assembled ONT data into contigs and filled gaps in the preliminary assembly with NextDenovo[58]. The assembled scaffolds were further polished using NextPolish (v.1.4.1)[59] software on Illumina and HiFi reads, and the polished scaffolds were used for gap filling with TGS-GapCloser (v.1.2.1)[60] and PGA (v.1.0)[61]. Finally, the six peanut genome assemblies were polished using T2T-polish (v.1.0)[62] with the automated-polishing option. After genome polishing, we generated six T2T peanut genomes.

### Identification of telomere and centromere regions

The VGP telomere pipeline (https://github.com/VGP/vgp-assembly) was used to identify telomeres for telomere determination. This involved a direct search for the telomere sequence 5′-CCCTAAA-3′ and its reverse complement. We used the CentroMiner module of QuarTeT (v.1.0.4)[63] to identify centromere regions for each chromosome. To validate the centromere regions, we performed chromosomal interaction strength analysis using HiC-Pro (v.2.11.1)[55]. In addition, we mapped the verified 116-bp repeat unit in peanut onto the genome using Lastz (v.1.04.22)[64] to identify telomere positions.

### Assembly assessment

For assembly assessment, genome accuracy was evaluated by aligning Illumina reads and the long reads (HiFi and ONT) to the six peanut genomes using BWA-MEM (v.0.7.17-r1188)[65] and minimap2 (v.2.24-r1122)[66]. The quality of the genome assembly was assessed by calculating the Merqury quality value using the Merqury 21-mer database, incorporating both Illumina PCR-free and PacBio HiFi reads. The completeness of the six genomes was assessed by BUSCO analysis with the embryophyta_odb10 database using BUSCO (v.5.4.3)[23]. Furthermore, LAI values were calculated for evaluation of the accuracy and completeness of LTR retrotransposons in the assembled genomes using LTR_FINDER (v.1.1)[67], ltrharvest (v.1.6.2)[68] and LTR_retriever (v.2.9.0)[69] in combination. Collinear regions and SVs were identified via NUCmer (v.3.1)[57] and SyRI (v.1.5.3)[26].

### TE annotation

RepeatModeler was used for de novo prediction of the repetitive elements of the genome assemblies, and genome TEs were identified with RepeatMasker (v.4.1.4)[70]. To precisely classify the LTR retrotransposons, TEsorter (v.1.4.6)[71] was used for further analysis of the LTR_retriever (v.2.9.0)[69] results. REXdb (http://repeatexplorer.org/) was used to improve the sensitivity of the results.

### Identification of genetic variants

To identify SVs, we generated two sets of SVs, calling results using SyRI (v.1.2)[26] and SVMU (v.0.4-alpha)[27], both with default parameters. For SV detection using SyRI (v.1.2)[26], pairwise genome alignments were first generated using minimap2 (v.2.21-r1071)[66] with the '-ax asm5-eqx' parameters. The resulting alignment files were then processed by SyRI (v.1.2)[26], with a minimum size threshold of 50 bp. For SV detection with SVMU (v.0.4-alpha)[27], intergenomic alignments were generated using the nucmer program in MUMmer4 (v.4.0.0beta2)[57] with the '−mum' parameter. Insertions and deletions larger than 50 bp inside syntenic alignment regions were also kept for further analysis. For CNVs, we applied a filter from the SVMU output based on length (>50 bp) and coverage criteria (reference or query coverage ≥2 or ≤0.5). Inversions greater than 1 Mb were extracted from the SyRI results and manually

verified. Finally, the identified SVs from each sample were merged using SURVIVOR (v.1.0.6)[72] with the following parameters: '50 1 0 0 0 0'.

## Gene prediction and functional annotation

A combination of ab initio, homology and RNA-seq prediction was used to annotate protein-coding genes. For ab initio gene prediction, Augustus (v.3.5.0)[73] and GlimmerHMM (v.3.0.4)[74] were used to construct the highest-confidence gene models. Protein sequences of *A. duranensis* and *A. ipaensis*[12], *A. hypogaea*[2], *Helianthus annuus* L.[75], *Glycine max*[76], *Brassica rapa* ssp.[77] and *Lupinus albus*[78] were used to predict homologous genes. Liftoff (v.1.6.3)[79] was used to annotate the candidate genome by referencing the published Tifrunner[2] genome and annotation files. For RNA-seq annotation, we sequenced various developmental stages using Illumina technology. For second-generation sequencing, reads were aligned to the reference genome assembly using HISAT2 (v.2.1.0)[80] with default parameters, BAM files were merged with SAMtools (v.1.16.1)[81], and transcripts were assembled with StringTie (v.2.2.1)[82] to produce GTF files. The longest transcripts were extracted using TransDecoder (v.5.5.0) software and compared to the UniProt_SwissProt database for optimal alignment to generate high-confidence GFF3 files. In addition, transcriptome data were de novo assembled using Trinity (v.2.1.1)[83]. The assembled transcripts were then aligned to the genome, and gene models were built using the PASA (v.2.5.2)[84] pipeline. Full-length transcriptome analysis was conducted using Iso-Seq3 (v.3.8.2) (https://github.com/PacificBiosciences/IsoSeq) and CD-Hit (v.4.8.1)[85] to perform transcript deduplication with parameters -c 0.90 -d 40 -M 256000 -U 10 -p 1 -G 1 -s 0.90. After completion of these steps, the generated gene models were integrated using EVidenceModeler (v.2.1.0)[86]. The weights assigned to each type of evidence were: Liftoff homology-set (weight: 10); ISO-set (weight: 10); TransDecoder (weight: 8); PASA (weight: 5); Ab-set (weight: 2); Homology-set (weight: 1). In addition, we enhanced gene function annotation by integrating data from the NCBI nonredundant protein database (https://ftp.ncbi.nlm.nih.gov/blast/db/FASTA/), Pfam database[87], InterProScan database[88] and Swiss-Prot database[89]. GO term annotations and protein domains were obtained using InterProScan[88], and KEGG annotations were performed with kofam_scan-1.3.0 using parameter -e '1e-5'.

## Resequencing and population structure analysis

A total of 521 accessions were selected for whole-genome resequencing, including 23 wild species, 131 landraces and 367 cultivated peanuts. The raw reads were filtered using Trimmomatic (v.0.36)[90] and then mapped to the reference genome (S245, silihong) using BWA-MEM (v.0.7.17-r1188)[65]. Variants were then identified using HaplotypeCaller and GenotypeGVCFs tools in the Genome Analysis Tool Kit (v.3.8)[91]. The obtained SNPs were filtered using the VariantFiltration option with 'QD < 2.0 || FS > 60.0 || MQ < 40.0 || MQRankSum < -12.5 || ReadPosRankSum < -8.0'. The filtered SNPs were used for the population structure analysis. We constructed a neighbor-joining phylogenetic tree using TreeBeST (v.1.9.2)[92] and visualized the tree topology using R package ggtree (v.3.4.4)[93]. PCA was conducted using PLINK (v.1.9)[94]. To estimate the ancestry proportions of each individual and quantify the genome-wide admixture, we used ADMIXTURE (v.1.3.0)[95] to infer $K = 2$ to $K = 15$ clusters of related individuals. The optimal number of groups was determined by calculating the cross-validation error. Nucleotide diversity ($\pi$) in 100-kb nonoverlapping sliding windows was calculated using VCFtools (v.0.1.17)[96]. The LD decay among the different groups was calculated using PopLDdecay (v.3.40)[97].

## Inference of demographic history by coalescent simulation

Combining the phylogenetic tree and population structure results, we inferred the demographic scenarios of the four cultivated groups using fastsimcoal2 (v.2.8)[98]. Four-fold degenerate synonymous sites (4DTv) were filtered using VCFtools (v.0.1.17)[96], and easySFS was used to convert the VCF data into various SFS formats. We then used these SFS data to fit five demographic models with options '-N 200,000 -m -0 -L 50 -M -q -s 0' and default settings. Wide search ranges with log-uniform distributions were used for parameter estimation, assuming a generation time of 1 year and a mutation rate of $1.64 \times 10^{-8}$ mutations per generation per site. Finally, 100 independent fastsimcoal2 (v.2.8)[98] runs were performed for each model to determine the parameter estimates leading to the maximum likelihood and to identify the best-fitting model.

## Selective sweep analysis

The $F_{ST}$ and XP-CLR test were employed to detect selective signatures between different groups. The $F_{ST}$ was calculated in 100-kb sliding windows with a step size of 10 kb using VCFtools (v.0.1.17)[96]. The XP-CLR (v.1.1.2)[99] method (with command line -w1 0.0005 100 100 1 -p 0 0.7) was used to scan each chromosome in 100-kb sliding windows with a step size of 10 kb. The top 1% of genomic regions with the highest scores in both the $F_{ST}$ and XP-CLR analyses were considered to be potential selection regions.

## Whole-genome analysis of genomic introgression

Genomic introgression was estimated at the population level. A four-taxon fd statistic[100] was used to identify genomic segments introgressed between different groups. In the absence of gene flow, the ABBA and BABA allele configurations in the four-taxon tree (((P1, P2), P3), O) would be expected to occur with equal frequency. Gene flow between groups would result in an excess of ABBA alleles relative to BABA alleles, detectable using the fd statistic. The fd statistics were calculated in 100-kb sliding windows with a step size of 10 kb using Python script ABBABABAwindows.py (https://github.com/simonhmartin/genomics_general). Windows with fewer than three informative SNPs (neither ABBA nor BABA) were excluded. Windows with negative Patterson's D-statistic values (closely related to the fd statistic) and those with fd > 1 were also excluded. The top 1% of introgressed regions were retained for further gene selection and enrichment analysis.

## GWAS analysis and candidate gene identification

To obtain accurate genetic phenotypes (Supplementary Table 32), we used an R (v.4.3.1) script in the lme4 package (https://cran.r-project.org/web/packages/lme4/) to calculate best linear unbiased prediction values, with location and year as random effects in the model: lmer (phenotypes~(1|loc) + (1|year) + (1|lines) + (1|year: lines). The best linear unbiased prediction values were used as individual phenotypes for GWAS analysis using the GAPIT (v.3.5)[101] R package. The first two principal components were regarded as covariates, and the kinship matrix was internally computed using GAPIT (v.3.5)[101]. The genome-wide significance threshold for association was set to $1/n$, where $n$ represents the number of markers. According to PopLDdecay (v.3.40)[97] results, only LD blocks containing at least one significant SNP and one suggested SNP were considered to be significant loci. Functional annotation was performed for each candidate gene associated with these significant loci, and homologs from the model species (rice) were identified using BLASTP. Gene haplotype analysis was conducted using population data, and the relationship between haplotypes and phenotypes was used to finalize the candidate genes.

## Metabolite profiling

Seed coat and kernel were harvested at five developmental stages (DAF15, DAF30, DAF45, DAF60 and DAF75) for the S245 and S83 accessions, with three biological replicates per stage. Samples were immediately frozen in liquid nitrogen and stored at −80 °C. Lipids were extracted using methanol and MTBE, and analyzed via liquid chromatography coupled with electrospray ionization tandem mass spectrometry with a Thermo Accucore C30 column. Anthocyanins were extracted with methanol/water/HCl and analyzed using ultra-performance liquid

chromatography coupled with electrospray ionization tandem mass spectrometry. Both lipid and anthocyanin contents were measured with MetWare on the AB SCIEX QTRAP 6500 platform.

## Weighted gene coexpression network analysis

For RNA-seq analysis, total RNA from seed coat and kernel at five developmental stages was extracted and subjected to quality control for library construction. Illumina reads were filtered to remove adaptor sequences, empty reads and low-quality reads ($Q < 20$) using FASTX-toolkit. Clean reads were aligned to the S245 reference genome with HISAT2 (v.2.1.0)[80]. Read counts were calculated using HTSeq (v.2.0.3)[102], and gene expression levels were estimated as FPKM values. WGCNA was performed using the WGCNA R package (v.1.72)[103], excluding genes with average expression less than 1. The expression matrix of metabolites was log-transformed for network analysis, with a soft thresholding power of 16 for peanut kernels and six for seed coats. Modules were identified and merged based on topological overlap. Genes were clustered into modules with a minimum size of 50, and modules were merged with a height threshold of 0.25 using the dynamic tree-cutting method. Relationships between modules and metabolite contents were assessed using module eigengenes and Pearson correlation coefficients. Differential metabolites were analyzed using correlation coefficients and $P$ values to select genes and metabolites with high expression correlation for further study.

## Subcellular localization

The CDSs of *AhWRI1* and *AhGSA1* without stop codons were cloned into the pCAMBIA1302 vector using a ClonExpress II One Step Cloning Kit (C112, Vazyme Biotech). The primers used for these assays are listed in Table 31. The recombinant constructs and the empty vector control were introduced into *Agrobacterium tumefaciens* strain GV3101 using the freeze-thaw method. *Agrobacterium* cultures were prepared and infiltrated into *Nicotiana* leaves. GFP fluorescence was observed 48 h after infiltration using a Nikon laser-scanning confocal microscope.

## Dual-luciferase and GUS assays

Promoter fragments of *AhWRI1* (350 bp upstream of ATG), *AhACP1* (265 bp), *AhKAS1* (312 bp) and *AhGSA1* (660 bp; Hap1 and Hap2 alleles) were amplified and cloned upstream of the firefly luciferase (LUC) gene in the reporter vector pGreenII 0800-LUC, which contains a 35S-driven Renilla luciferase (REN) cassette as an internal control. For the SNP assay, a 52-bp fragment spanning the C/A polymorphism in the *AhWRI1* promoter was similarly cloned into pGreenII 0800-LUC. Full-length CDSs of *AhFUS3*, *AhABI3*, *AhLEC1* and *AhWRI1* were cloned into effector vector pGreenII 62-SK under the CaMV 35S promoter. Reporter and effector constructs were transformed into *A. tumefaciens* GV3101, adjusted to an optical density at 600 nm of 0.6–0.8, and coinfiltrated into *Nicotiana benthamiana* leaves at a 1:1 (v/v) ratio. Leaves were harvested 48 h after infiltration, and LUC and REN activities were measured using a Dual-Luciferase Reporter Assay Kit (Beyotime). Promoter activity was expressed as the LUC/REN ratio. For GUS assays, the Hap1 and Hap2 *AhGSA1* promoter fragments (660 bp upstream of ATG) were cloned into GUS reporter vector pCAMBIA1381, and GUS activity was assessed by histochemical staining. The relevant primers are shown in Table 31.

## Real-time qPCR Assay

qPCR reactions were performed on a QuantStudio 5 (Thermo Fisher Scientific) using Hieff qPCR SYBR Green Master Mix (Low Rox Plus, 11202ES03). The primers used for the RT–qPCR assays are listed in Table 31. The thermal cycling conditions were as follows: initial denaturation at 95 °C for 10 min, followed by 40 cycles of 95 °C for 15 s and 60 °C for 1 min. Melt curve analysis was conducted to verify the specificity of the amplification; three biological replicates were set for each gene. Relative gene expression levels were calculated using the $2^{-\Delta\Delta Ct}$ method.

## Yeast one-hybrid analysis

Protein–DNA interactions were tested using a Matchmaker Gold Yeast One-Hybrid System (Takara Bio/Clontech, catalogue number 630491). Promoter fragments of *AhWRI1*, *AhACP1* and *AhKAS1* (350 bp, 265 bp and 312 bp upstream of ATG, respectively) were cloned into the pAbAi vector and integrated into the genome of the Y1HGold yeast strain to generate bait reporter strains. Each bait strain was tested for self-activation, and the minimal inhibitory concentration of aureobasidin A was determined. Full-length CDSs of *AhFUS3*, *AhABI3*, *AhLEC1* and *AhWRI1* were cloned into the pGADT7-AD vector (prey). The relevant primers are shown in Table 31. Prey plasmids were transformed into the corresponding bait strains, and transformants were selected on SD/−Leu medium supplemented with aureobasidin A. Colony growth on selective plates indicated a positive interaction.

## Transmission electron microscopy sample preparation

Leaf tissues were vacuum infiltrated with glutaraldehyde fixative for 1 h and fixed overnight at 4 °C. Samples were washed 3 times in 0.1 M phosphate buffer (15 min each), post-fixed in osmium tetroxide for 4 h, washed in phosphate buffer 3 times (15 min each) and stained with 1% uranyl acetate for 1 h. Samples were then dehydrated through a graded ethanol series (30%, 50%, 70%, 90% and 100%; 25 min each) and incubated in propylene oxide for 1 h. Tissues were infiltrated with a graded resin series (3 changes, 4 h each), followed by 2 changes of pure resin (8 h each), and polymerized at 60 °C for 48 h.

## Nile red staining

After *Agrobacterium* infiltration, rapeseed and soybean leaf discs were collected and fixed with paraformaldehyde in 1× phosphate-buffered saline. Nile red (dissolved in dimethyl sulfoxide) was diluted in 1× phosphate-buffered saline to a final concentration of 4 μg ml$^{-1}$ for lipid droplet staining. Leaf discs were imaged immediately using a Leica TCS SP8 confocal fluorescence microscope. The excitation wavelength was 488 nm, and emission was collected at 560–620 nm. Images were acquired as z-stacks comprising 10 optical sections. The field of view was 161.7 × 161.7 μm. Lipid droplets were quantified using ImageJ, and counts were normalized to area (number of LDs per mm$^2$). Images were obtained from three biological replicates.

## Gravimetric lipid determination

Briefly, 25 mg of dried mature seeds were weighed to ±0.01 mg on a semimicroanalytical balance (Mettler Toledo) and ground in 0.5 ml of hexane/isopropanol (6:4, v/v); the extract was brought to 2 ml with the same solvent. Samples were vortexed for 2 min and sonicated for 15 min at room temperature. An aqueous sodium sulfate solution (1 ml) was added, and samples were vortexed and centrifuged at 1,500$g$ for 5 min to separate phases. The lower phase was reextracted 3 times with 2 ml of hexane/isopropanol (6:4, v/v). Upper phases were combined, transferred to clean glass tubes and evaporated under oxygen-free nitrogen to constant weight. Oil content was calculated as the extracted lipid mass divided by the initial seed dry weight. For each genotype, three biological replicates were analyzed.

## Statistical analyses

Statistical analyses were performed in R (v.4.1.3) using two-tailed Student's *t*-tests. For the biochemical and molecular biology analyses, at least three biological replicates were used for each sample.

## Reporting summary

Further information on research design is available in the Nature Portfolio Reporting Summary linked to this article.

## Data availability

The genome assembly data for this project have been deposited in the National Center for Biotechnology Information (NCBI) database

under BioProject accession code PRJNA1259200 and with the National Genomics Data Center, Beijing Institute of Genomics, Chinese Academy of Sciences, under BioProject accession code PRJCA026588. The genome assembly and annotation are also available from the Peanut Genome Database (http://omicsplant.cn/peanut) hosted by the Peking University Institute of Advanced Agricultural Sciences and via figshare at https://figshare.com/articles/dataset/T2T_genome_assembly_in_peanut/28883636 (ref. 104). All raw sequencing data for the genome assemblies and annotations, transcriptome and PacBio Iso-Seq reads, and whole-genome sequence data for the population and GWAS analyses have been deposited in the NCBI database under BioProject accession code PRJNA1259747. Source data are provided with this paper.

## Code availability

No custom code or mathematical algorithm was used in this work.

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

## Acknowledgements

This study was supported by the Key R&D Program of Shandong Province, China (2024LZGC035, ZR202211070163); the Taishan Scholars Program of Shandong Province (tsqn202103161); the Natural Science Foundation of Shandong Province (ZR202103010405); the Weifang Science and technology development plan (2024JZ001); and the Director's Award from the Peking University Institute of Advanced Agricultural Sciences to X.L. The study was also supported by awards from the Natural Science Foundation of Shandong Province (ZR2022ZD22), the National Top Young Talents Program of China, and the National Key R&D Program of China (2019YFD1000200) to H.H.; from the Key R&D Program of Shandong Province, China (ZR202211070163) to X.W.D.; from the Shandong Provincial Science and Technology Innovation Fund. R.K.V. thanks the Food Futures Institute of Murdoch University for a start-up grant. We also thank the High-Performance Computing center of Peking University Institute of Advanced Agricultural Science, Shandong Laboratory for Advanced Agricultural Sciences for providing computational resources in this work.

## Author contributions

X.L., H.H. and R.K.V. conceived and designed the project. J.B., K.L. and Y.Z. performed the genome assembly. J.B. and S.D. contributed to gene annotation, repeat annotation and repeat evolution and performed centromere and telomere analyses. J.B., X.L., G.Z. and Y.L. performed the population and GWAS analyses. Y.G., J.L., Y.C., Q.M., H.G. and T.L. performed validation of SVs and functional genes. X.L., H.G., J.L., Y.G., L.H., C.Z., X.W., R.T., V.G. and R.K.V. prepared the plant materials and generated the sequencing data. Y.Z., J.B. and K.L. performed metabolic analyses. J.B., S.D. and H.G. contributed to data management. J.B., X.L., H.H., Y.Z. and K.L. drafted the paper, and X.L., H.H., J.B., R.K.V., X.W.D., V.G., A.C. and L.S.Z. revised it. All authors read and approved the final version of the paper.

## Competing interests

The authors declare no competing interests.

## Additional information

**Correspondence and requests for materials** should be addressed to Rajeev K. Varshney, Hang He or Xiaoqin Liu.

# Reporting Summary

## Statistics

For all statistical analyses, confirm that the following items are present in the figure legend, table legend, main text, or Methods section.

| n/a | Confirmed | |
|---|---|---|
| ☐ | ☒ | The exact sample size (*n*) for each experimental group/condition, given as a discrete number and unit of measurement |
| ☐ | ☒ | A statement on whether measurements were taken from distinct samples or whether the same sample was measured repeatedly |
| ☐ | ☒ | The statistical test(s) used AND whether they are one- or two-sided *Only common tests should be described solely by name; describe more complex techniques in the Methods section.* |
| ☒ | ☐ | A description of all covariates tested |
| ☐ | ☒ | A description of any assumptions or corrections, such as tests of normality and adjustment for multiple comparisons |
| ☐ | ☒ | A full description of the statistical parameters including central tendency (e.g. means) or other basic estimates (e.g. regression coefficient) AND variation (e.g. standard deviation) or associated estimates of uncertainty (e.g. confidence intervals) |
| ☐ | ☒ | For null hypothesis testing, the test statistic (e.g. *F*, *t*, *r*) with confidence intervals, effect sizes, degrees of freedom and *P* value noted *Give P values as exact values whenever suitable.* |
| ☒ | ☐ | For Bayesian analysis, information on the choice of priors and Markov chain Monte Carlo settings |
| ☒ | ☐ | For hierarchical and complex designs, identification of the appropriate level for tests and full reporting of outcomes |
| ☒ | ☐ | Estimates of effect sizes (e.g. Cohen's *d*, Pearson's *r*), indicating how they were calculated |

*Our web collection on statistics for biologists contains articles on many of the points above.*

## Software and code

Policy information about availability of computer code

| | |
|---|---|
| Data collection | The resequencing data was collected using the DNBSEQ-T7 platform. HIFI data was collected using the PacBio Sequel II sequencing platform (Pacific Biosciences, CA, USA). ONT data was collected using a PromethION sequencer. Hi-C data was collected using the Illumina NovaSeq 6000 platform. RNA date was collected using Illumina and Sequel II/IIe system (Pacific Biosciences, CA, USA) sequencing platform. |
| Data analysis | These software include fastp (v.0.22.0); HiC-Pro (v.2.11.1); Hifiasm (v.0.19.5); NUCmer (v.3.1); NextDenovo (v.2.5.0); NextPolish (v.1.4.1); TGS-GapCloser (v.1.2.1); PGA (v.1.0); T2T2-polish (v.1.0); QuarTeT (v.1.0.4); BWA-MEM (v.0.7.17-r1188); minimap2 (v.2.24-r1122); SyRI (v.1.5.3); SVMU (v.4.0.0beta2); Merqury; BUSCO (v.5.4.3); TEsorter (v.1.4.6); MUMmer4(v.4.0.0beta2); Augustus (v.3.5.0); GlimmerHMM (v.3.0.4); Liftoff (v.1.6.3); SAMtools (v.1.16.1); TransDecoder (v.5.5.0); Trinity (v.2.1.1); LTR_FINDER (v1.1); ltrharvest (v.1.6.2); LTR_retriever (v.2.9.0); CD-Hit (v.4.8.1); EVidenceModeler (v.2.1.0); JCVI (v.1.3.4); ParaAT(v.2.0); KaKs_Calculator (v.3.0); TreeBeST (v.1.9.2); PLINK (v.1.9); ADMIXTURE (v.1.3.0); VCFtools (v.0.1.17); PopLDdecay (v.3.40); fastsimcoal2 (v.2.8); XP-CLR (v.1.1.2); R (v.4.3.1); "lme4" package (https://cran.r-project.org/web/packages/lme4/); GAPIT (v.3.5); Stringtie2 (v.2.1.7) ; RepeatMasker(v.4.1.4); HISAT2 (v.2.1.0); HTSeq (v.2.0.3); WGCNA R package (v.1.72); Lastz(v.1.04.22); Genome Analysis Tool Kit (GATK) (v.3.8); Trimmomatic (v.0.36);SVMU (v. 0.4-alpha) |

For manuscripts utilizing custom algorithms or software that are central to the research but not yet described in published literature, software must be made available to editors and reviewers. We strongly encourage code deposition in a community repository (e.g. GitHub). See the Nature Portfolio guidelines for submitting code & software for further information.

## Data

The genome assembly for this project have also been deposited in the National Center for Biotechnology Information (NCBI) database under BioProject number PRJNA1259200 and the National Genomics Data Center, Beijing Institute of Genomics, Chinese Academy of Sciences, under BioProject accession no. PRJCA026588. Additionally, the genome assembly and annotation are also available in the Peanut Genome Database (http://omicsplant.cn/peanut) hosted by the Peking University Institute of Advanced Agricultural Sciences and Figshare (https://figshare.com/articles/dataset/T2T_genome_assembly_in_peanut/28883636). All raw sequencing data for the genome assemblies and annotations, transcriptome and PacBio Iso-Seq reads, and whole-genome sequence data for the population and GWAS analyses have been deposited in the National Center for Biotechnology Information (NCBI) database under BioProject number PRJNA1259747.

## Research involving human participants, their data, or biological material

| | |
|---|---|
| Reporting on sex and gender | N/A |
| Reporting on race, ethnicity, or other socially relevant groupings | N/A |
| Population characteristics | N/A |
| Recruitment | N/A |
| Ethics oversight | N/A |

Note that full information on the approval of the study protocol must also be provided in the manuscript.

# Field-specific reporting

Please select the one below that is the best fit for your research. If you are not sure, read the appropriate sections before making your selection.

☒ Life sciences    ☐ Behavioural & social sciences    ☐ Ecological, evolutionary & environmental sciences

For a reference copy of the document with all sections, see nature.com/documents/nr-reporting-summary-flat.pdf

# Life sciences study design

All studies must disclose on these points even when the disclosure is negative.

| | |
|---|---|
| Sample size | We used 6 peanut accessions for genome assembly and 521 accessions collected worldwide for population genetics analysis. |
| Data exclusions | No data was exclude. |
| Replication | Three biologically independent samples were used for transcription and qRT-PCR analysis. |
| Randomization | A randomized complete block design was used in planting for phenotype data collected in four growing seasons. |
| Blinding | All accessions were only labeled by numbers when planting and data collection. |

# Reporting for specific materials, systems and methods

We require information from authors about some types of materials, experimental systems and methods used in many studies. Here, indicate whether each material, system or method listed is relevant to your study. If you are not sure if a list item applies to your research, read the appropriate section before selecting a response.

## Materials & experimental systems

| n/a | Involved in the study |
|-----|----------------------|
| ☒ ☐ | Antibodies |
| ☒ ☐ | Eukaryotic cell lines |
| ☒ ☐ | Palaeontology and archaeology |
| ☒ ☐ | Animals and other organisms |
| ☒ ☐ | Clinical data |
| ☒ ☐ | Dual use research of concern |
| ☐ ☒ | Plants |

## Methods

| n/a | Involved in the study |
|-----|----------------------|
| ☒ ☐ | ChIP-seq |
| ☒ ☐ | Flow cytometry |
| ☒ ☐ | MRI-based neuroimaging |

# Dual use research of concern

Policy information about dual use research of concern

## Hazards

Could the accidental, deliberate or reckless misuse of agents or technologies generated in the work, or the application of information presented in the manuscript, pose a threat to:

| No | Yes |
|----|-----|
| ☒ ☐ | Public health |
| ☒ ☐ | National security |
| ☒ ☐ | Crops and/or livestock |
| ☒ ☐ | Ecosystems |
| ☒ ☐ | Any other significant area |

## Experiments of concern

Does the work involve any of these experiments of concern:

| No | Yes |
|----|-----|
| ☒ ☐ | Demonstrate how to render a vaccine ineffective |
| ☒ ☐ | Confer resistance to therapeutically useful antibiotics or antiviral agents |
| ☒ ☐ | Enhance the virulence of a pathogen or render a nonpathogen virulent |
| ☒ ☐ | Increase transmissibility of a pathogen |
| ☒ ☐ | Alter the host range of a pathogen |
| ☒ ☐ | Enable evasion of diagnostic/detection modalities |
| ☒ ☐ | Enable the weaponization of a biological agent or toxin |
| ☒ ☐ | Any other potentially harmful combination of experiments and agents |

# Plants

| | |
|---|---|
| Seed stocks | *Report on the source of all seed stocks or other plant material used. If applicable, state the seed stock centre and catalogue number. If plant specimens were collected from the field, describe the collection location, date and sampling procedures.* |
| Novel plant genotypes | *Describe the methods by which all novel plant genotypes were produced. This includes those generated by transgenic approaches, gene editing, chemical/radiation-based mutagenesis and hybridization. For transgenic lines, describe the transformation method, the number of independent lines analyzed and the generation upon which experiments were performed. For gene-edited lines, describe the editor used, the endogenous sequence targeted for editing, the targeting guide RNA sequence (if applicable) and how the editor was applied.* |
| Authentication | *Describe any authentication procedures for each seed stock used or novel genotype generated. Describe any experiments used to assess the effect of a mutation and, where applicable, how potential secondary effects (e.g. second site T-DNA insertions, mosiacism, off-target gene editing) were examined.* |

