## [Peer Review File · Nature Genetics]

Telomere-to-telomere genome assemblies and population resequencing of diploid and allotetraploid peanut varieties

Corresponding Author: Ms Xiaoqin Liu

Version 0:

Decision Letter:

25th Jun 2024

Dear Professor Liu,

Your Article, "Origin, evolution and domestication of cultivated peanuts" has now been seen by 3 referees. You will see from their comments copied below that while they find your work of considerable potential interest, they have raised quite substantial concerns that must be addressed. In light of these comments, we cannot accept the manuscript for publication, but would be very interested in considering a revised version that addresses these serious concerns.

We hope you will find the referees' comments useful as you decide how to proceed. If you wish to submit a substantially revised manuscript, please bear in mind that we will be reluctant to approach the referees again in the absence of major revisions.

To guide the scope of the revisions, the editors discuss the referee reports in detail within the team, including with the chief editor, with a view to identifying key priorities that should be addressed in revision and sometimes overruling referee requests that are deemed beyond the scope of the current study. In this case, we ask you to address the reviewers' comments in full. We expect you to address all the technical concerns raised by the Reviewers, especially those regarding the analysis of the estimated age of polyploidy (Reviewer #2) and the potential confounding of population structure for the GWAS results (Reviewer #1).

If you choose to revise your manuscript taking into account all reviewer and editor comments, please highlight all changes in the manuscript text file. At this stage we will need you to upload a copy of the manuscript in MS Word .docx or similar editable format.

*2) If you have not done so already please begin to revise your manuscript so that it conforms to our Article format instructions, available here. Refer also to any guidelines provided in this letter.

*3) Include a revised version of any required Reporting Summary: <https://www.nature.com/documents/nr-reporting-summary.pdf>

Please be aware of our guidelines on digital image standards.

Link Redacted

If you wish to submit a suitably revised manuscript we would hope to receive it within 6 months. If you cannot send it within this time, please let us know. We will be happy to consider your revision so long as nothing similar has been accepted for publication at Nature Genetics or published elsewhere. Should your manuscript be substantially delayed without notifying us in advance and your article is eventually published, the received date would be that of the revised, not the original, version.

Thank you for the opportunity to review your work.

Sincerely,
Chiara

Chiara Anania, PhD
Associate Editor
Nature Genetics
<https://orcid.org/0000-0003-1549-4157>

Referee expertise:

Referee #1: plant biology; evolutionary genomics

Referee #2: peanut genomics; evo

Referee #3: signed review

Reviewers' Comments:

Reviewer #1:

Remarks to the Author:

This is an interesting paper, but I find some important attention to detail is missing, and that several findings need better couching as speculative or even whether they're based on a priori evidence. Some points to consider, in no particular order:

- I suppose the major significance of the TE analyses is timing of bloom vs. tetraploidization, one needs to consider mutation rates used for Ks versus TEs.. the same or different? One could anticipate that both should be proxies for base neutral mutation rate. For example, do the the different species have different generation times?
- Differences in centromeric repeat array lengths. The authors suggest "significant differentiation of centromeric regions after tetraploidization." But they have sequenced modern exemplars of parental sub genomes, so it remains unknown what the arrays were like in the true ancestral genomes. I would rephrase.
- SVs - I strongly suggest that these be binned by molecular mechanism, so that the results make better biological sense (even though much of the literature conflates them). So what about indels, CNVs (tandems), TE insertions/ectopic recombinations, etc. The above can become important for functional annotation of SVs. For example, tandems (and new or sub- functional potential) can have very different functional relevance than indels, which can even include TE or NUMT NUPT insertions into noncoding regulatory sequence. And of course, TEs hopping into exons and introns can be a different functional class to consider.
- Note that ADMIXTURE will always return K ancestral populations, even when one K is preferred ad hoc or by cross-validation (eg). Thus the results need to be couching in this respect; they are not necessarily "real". It's good that the K separates phylogenetic lineages - highlight this better.
- Selective sweeps. Recent TE or NUMT/NUPT insertions can fool these analyses by presenting highly homozygous

stretches of the genome. Look through your results for these molecular phenomena - at least using a sanity check of about 100 random hits. You may wish to filter out "sweep" hits that better reflect TE or NUPT/NUMT homozygosity before functional enrichment analysis. Also, what happens in terms of enrichment when you consider top 1% instead of 5%? More conservative could be safer

- GWAS: how did you convincingly control for what must be considerable underlying population structure in your sample? For example, among the different phylogenetic lineages.
- Were the candidate nut phenotype genes identified a priori, or solely through anonymous bioinformatics? If the former, and corroborated informatically, admit that please explicitly ;)
- WGD analyses. Please use syntenic analysis, not only Ks, which can be sensitive to underlying demographic differences, eg
- Functional enrichments - what were the exact backgrounds used? They should be included as files for replicability
- Phylogenetic trees should either display support values or parenthetical tree files should be provided in supplemental so investigators can evaluate robustness of results
- Fig. 4B shows a pretty clear horizontal cline. Carefully evaluate how this affected GWAS. And does the PCA suggest admixed populations (intermediate between red and green on X axis) in similar way as your other analyses?
- Where is the TreeMix analysis? Where are the f stats in tabular form? What were the significance values used. Etc.

Reviewer #2:

Remarks to the Author:

The study concerns a large sequencing analysis of peanuts, with several complete genomes and hundreds of peanut accessions characterized by whole genome sequencing. One major problem is that one of the basal analysis on the estimated age of polyploidy is fundamentally flawed, leading to wrong conclusions propagating through the manuscript:

The authors infer the age of divergence of two genomes from peaks in Ks curves constructed using the WGD software. The WGD software is constructed to find evidence for ancient Whole Genome Duplications, by intragenomic (not intergenomic) comparison. When making an intragenomic comparison (a comparison of a genome with itself), it's clear that there will always be 100% self-identity. Since the software is searching for residual, much weaker similarities from a WGD, it filters Ks values at (or close to) zero. However, when making comparisons between two different genomes, there is no self-identity to eliminate. Each gene should be compared to its best match in the other genome (a comparison of orthologs - the authors use the word "ortholog", but in fact, orthologous comparisons are filtered out by the software they use).

The authors kindly added several self-comparisons to their figure. These new self-comparison curves serve as controls for the method. Considering the objectives of the authors, these control curves should give values of zero divergence because the genomes are identical to themselves. In fact, none of the self-comparisons have a peak at zero. The methods used by the author are unable to show that a genome has zero divergence from itself!

In the same way, the method is unable to give a correct divergence of two genomes that are very similar to each other. Reiterating, this is because the method is designed for self-comparisons of a polyploid genome to indicate the age of divergence of the subgenomes. Values at (or close to) zero are filtered out. This is the reason the peak in the S245 (Ahy) vs S245 (Ahy) curve is Ks 0.025 (=1.58 Mya -the age divergence of A and B subgenomes), instead of zero.

For instance, the genome of Y4 (*A. ipaensis*) is, in fact, remarkably similar to the B subgenome of *A. hypogaea*. From the author's Ks data (S83B.cds.fa_Y4.cds.fa.ks.tsv) 27,923 genes are compared between S83 and Y4. Of these 24,541 have a Ks of zero. Indeed 23,584 of the genes have no nucleotide differences at all in their coding regions (Ks and Ka are both zero). The filtering of Ks=0 values combined with some sort of curve regression results in completely wrong Ks=0 values being represented on the graph.

The authors did not rescale the vertical axis as requested (It was relabelled as "Frequency", but the curves remain the same). If a number of gene pairs (=frequency) had been used, it would have been possible to see that data had gone missing from the different areas under the different curves.

I have attached three pdfs to help explain more:

Reviewer #3:

Remarks to the Author:

Overall evaluation

=====

This manuscript presents the sequencing, assembly and analysis of six peanut genomes. It expands the analysis with an

association analysis between two traits, oil content and seed weight and a genotype collection of 521 accessions. The manuscript is clear and easy to follow, although it contains a lot of information of results representing an incredible amount of work. The material and methods and the discussion should be improved in order to have a better support of the results presented. Overall the manuscript is well built, has interest and present relevant results not only for the peanut scientific community but also for the genomic community working in genome evolution and polyploidy.

Point-By-Point Manuscript Evaluation:

A. Summary of the key results

The manuscript titled "Origin, evolution and domestication of cultivated peanuts" describes the evolution of tetraploid peanut (*Arachis hypogaea* L.) genome through the sequencing and analysis of two diploid, and four tetraploid genomes. It also identifies genomic regions associated to domestication and to several agronomical traits through the re-sequencing of more than 500 accessions. Finally, the authors analyze the transcriptomics and metabolomics networks of the different peanut domesticated lineages. The results are summarized in the following points:

- Sequencing, assembly, and annotation of six representative accessions of peanuts, two diploids' progenitors (*A. duranensis*, AA, Y26 and, *A. ipaensis*, BB, Y4) and four tetraploids (*A. hypogaea*, var. *fastigiata*, var. *vulgaris*, var. *hypogaea*, var. *hirsute*). All the genomes were sequenced with PacBio HiFi, scaffolded with Hi-C data and the gaps filled with ONT. Genome size ranged from 1.18 Gb to 2.62 Gb. Each of the six assemblies presented excellent assembly metrics in terms of Illumina mapping rate (>99%), BUSCO completeness (>95%), LAI (>20), and Merqury QV (>40). The genome annotation produced 34-35K gene models for the diploid accessions and 71-72K for the tetraploids.
- TE analysis reported a range from 75 to 78% of repetitive elements. B subgenomes have a higher TE content than A being the Gypsy elements the main contributors. Age divergency between both subgenomes was dated 1.58 MYA, meanwhile the divergency between the AA-At and BB-Bt subgenomes was estimated 0.27 and 0.29 MYA in agreement with previously published results. A deeper TE analysis revealed different insertion patterns for the LTR elements after the polyploidization event. Centromeric regions were different between the diploid and the tetraploid accessions. Retand elements expanded after the polyploidization event contributed to the differentiation between AA and At subgenomes.
- The analysis of the SVs between the A, B sub-genomes and their ancestors revealed a higher number of inversions and translocations in the A sub-genome, compared with the B. Genes closed to the SVs were enriched in functions such as flavonoid biosynthesis, brassinosteroid biosynthesis, fatty acid metabolism and linoleic acid metabolism. Comparisons across different *A. hypogaea* subspecies revealed also a different functional enrichment with could contribute to the different phenotypes in some lineages like *hirsuta*.
- Population structure analysis with 521 peanut accessions revealed 5 distinct groups. Gene flow was detected between two different groups: G2-G3 and G4-G5.
- GWAS analysis on peanut oil content reported a significant signal on chromosome 8 with 151 genes. A gene named AhWRI1 was identified as responsible for the trait. Two haplotypes were identified in the collection with different oil content (~48% hap 1 and ~54% hap 2) associated with DE. Overexpression of this gene on rapeseed and soybean lines produced accumulation of lipid droplets supporting the involvement of this gene in the peanut oil content trait. Y1H assays allowed to associate several genes such as AhFUS3, AhABI3 and AhLEC1 to the regulation of the expression of this gene. A mutation on the E-Box promoter of this gene was identified as responsible for the different interaction with the AhFUS3 gene.
- GWAS analysis on seed size and weight revealed a peak on chromosome 16 spanning 130 genes. A gene named GSA1 was identified as one of the possible candidate genes for this trait based on similar phenotypes in rice. Two different haplotypes were identified for this gene, hap1 with 43 accessions and an average grain size of 650 gr. and hap2 with 294 accessions and an average grain size of 920 gr. The association of this candidate gene with the trait was verified with transcriptomic data as well as some analysis on the promotor of the gene.
- Transcriptomics and metabolomics comparisons between the accessions S83 and S245 identified key genes involved on lipid and anthocyanin content.

B. Originality and significance: if not novel, please include reference

The manuscript is original and novel. It expands our knowledge about the peanut domestication, and it generates important data for the community. It also describes two genes associated to two important agronomical traits.

C. Data & methodology: validity of approach, quality of data, quality of presentation.

The methodology is valid, and the quality of the presentation is good. More details in the material and methods are needed, specially to assess the RNA-Seq and the GWAS analysis that it has been evaluated under the principles the it has the right number of replicates.

D. Appropriate use of statistics and treatment of uncertainties

It is not applicable to this type of research, except for the GWAS and the RNA-Seq which looks fine. Nevertheless more details are needed in the M&M for a full assessment.

E. Conclusions: robustness, validity, reliability

Most of the conclusions are robust and valid although the lack of some information on the M&M (e.g., replicates) makes difficult to have a full assessment.

Dear Professor Liu,

Your Article, "Origin, evolution and domestication of cultivated peanuts" has now been seen by 3 referees. You will see from their comments below that while they find your work of interest, some important points are raised. We are interested in the possibility of publishing your study in Nature Genetics, but would like to consider your response to these concerns in the form of a revised manuscript before we make a final decision on publication.

To guide the scope of the revisions, the editors discuss the referee reports in detail within the team, including with the chief editor, with a view to identifying key priorities that should be addressed in revision and sometimes overruling referee requests that are deemed beyond the scope of the current study. In this case, we ask you to take seriously into account Reviewer #2 concerns and address all the remaining requests. We hope that you will find the prioritized set of referee points to be useful when revising your study. Please do not hesitate to get in touch if you would like to discuss these issues further.

We therefore invite you to revise your manuscript taking into account all reviewer and editor comments. Please highlight all changes in the manuscript text file. At this stage we will need you to upload a copy of the manuscript in MS Word .docx or similar editable format.

*2) If you have not done so already please begin to revise your manuscript so that it conforms to our Article format instructions, available

http://www.nature.com/ng/authors/article_types/index.html here

*3) Include a revised version of any required Reporting Summary: <https://www.nature.com/documents/nr-reporting-summary.pdf>

Please be aware of our <https://www.nature.com/nature-research/editorial-policies/image-integrity> guidelines on digital image standards.

EXTENDED DATA FIGURES

Link Redacted

We hope to receive your revised manuscript within four to eight weeks. If you cannot send it within this time, please let us know.

Nature Genetics is committed to improving transparency in authorship. As part of our efforts in this direction, we are now requesting that all authors identified as 'corresponding author' on published papers create and link their Open Researcher and Contributor Identifier (ORCID) with their account on the Manuscript Tracking System (MTS), prior to acceptance. ORCID helps the scientific community achieve unambiguous attribution of all scholarly contributions. You can create and link your ORCID from the home page of the MTS by clicking on 'Modify my Springer Nature account'. For more information please

visit please visit www.springernature.com/orcid.

Sincerely,
Chiara

Chiara Anania, PhD
Associate Editor
Nature Genetics
<https://orcid.org/0000-0003-1549-4157>

Referee expertise:

Referee #1:

Referee #2:

Referee #3:

Reviewers' Comments:

Reviewer #1 (Remarks to the Author):

After enormous revisions, many of which entailed significant new bioinformatic research, this manuscript is now an excellent example of a paper with the interest level and attention to detail requisite for a journal such as Nature Genetics. I am satisfied with how the investigators followed up on my suggestions, and am also particularly impressed at how independent approaches (fastsimcoal being one of them) yielded similar dates. This consistency speaks to both care taken to analyses as well as, of course, the appropriateness of the data to address the questions at hand. My only final suggestion - which I think is very important these days - is to revise all main-text figures such that they use similar (and best, color-blind compatible) color palettes. For example, Figs. 1 and 2 stand out very starkly in "standard" red, blue, and (bright) green colors, while Figs. 3 and 4, e.g., have much more muted colors that are easier to view.

Reviewer #2 (Remarks to the Author):

The following conclusions of the abstract aren't supported:

Lines 42-43

"a 13.23 Mb-long insertion on Chr14 of all the var. hirsute accessions may have contributed to the prostrate growth and narrow leaves of this group"

The data in the paper shows correlation - not causation

Lines 43-44

"Selective sweep and introgression analyses highlighted multiple asymmetric selections between the two subgenomes during breeding improvement."

There is no data to show this. Differences between the subgenomes were almost all accumulated in the diploid state, before the origin of *Arachis hypogaea*

Lines 45-46

"Genome-wide association studies identified 4,638 loci associated with 37 agronomic traits, including candidate genes for oil content, seed size, seed weight, etc."

Almost all of these loci lack any sort of verification.

Lines 50-51

"These findings provide a comprehensive understanding of the origin, evolution and domestication"

It's handwaving - there are few new insights. From this paper, I've learned nothing new of the origin, a bit about the evolution of the tetraploid genome (mainly about some rearrangements), and nothing at all about domestication.

More detailed comments

The authors shouldn't use the abbreviations Y26 and Y4 and instead use the full species names along with their original collection numbers (e.g., *Arachis ipaensis* K30076). This practice is crucial for properly acknowledging prior scientific work and enabling the stepwise accumulation of knowledge within the field.

accumulated any. Having said this, the best way to date close divergences like this is SNP rates from, whole genome alignments.

Why does Supplementary Figure 14 (B) use two different methods for the divergence dates K_s for the origin and coalescent-based genetic simulation for the others? The latter is giving dates I would expect (although they are stated with absurd precision (eg 9,491 years).

Reviewer #3 (Remarks to the Author):

Overall evaluation on the revised version

=====
The authors have successfully resolved the concerns and questions risen during the first revision. The manuscript, although it contains a huge amount of information, is clear and easy to read. The results are well supported by several data sources (phenome2genome associations, transcriptomics...). As I mentioned in my first revision, the manuscript is well built, has interest and present relevant results not only for the peanut scientific community but also for the genomic community working in genome evolution and polyploidy.

Revised by Aureliano Bombarely on December 23rd, 2024

Version 2:

Decision Letter:

31st Mar 2025

Dear Professor Liu,

Your Article entitled "Telomere-to-telomere genomes of diploids and allotetraploids provide insights into the origin, evolution, and domestication of peanuts" has now been seen by 3 referees, whose comments are attached. In the light of these advice we have decided that we cannot offer to publish your manuscript in Nature Genetics.

We feel that the ongoing technical reservations related to the incomplete assembling of some chromosomes are sufficiently important as to preclude publication of this study in Nature Genetics.

I am sorry that we cannot be more positive on this occasion, but hope that you will find our referees' comments helpful when preparing your paper for submission elsewhere.

Thank you.

Sincerely,
Chiara

Chiara Anania, PhD
Associate Editor
Nature Genetics
<https://orcid.org/0000-0003-1549-4157>

Reviewers' Comments:

Reviewer #1 (Remarks to the Author):

I am entirely satisfied with the authors' acceptance of my suggestion re: colorblind palettes:

"Regarding your final suggestion about figure color palettes, we fully agree with your point about the importance of using color-blind friendly palettes. In the current revised manuscript, we have ensured that all figures adopt consistent and more accessible color schemes, with careful attention to color-blind compatibility. We have updated Fig. 1 and Fig. 2 to replace the bright red, blue, and green with more muted, color-blind friendly alternatives, in line with the colors used in Figs. 3 and 4."

Regarding their responses to the other reviewers, in a short look, they seem well-considered, but I'm afraid I don't have time to carefully evaluate them.

GWAS Analysis:

In addition to the known loci (FAD and AhRUVBL2), the authors are encouraged to include genetic mechanism analyses for one or two additional phenotypic traits to enhance the GWAS findings.

Title & Writing Style:

The current title is too informal, and the overall writing style lacks academic rigor (e.g., the use of "etc." in the Abstract). A more precise and professional revision should be recommended.

Version 5:

Decision Letter:

Our ref: NG-A65542R4

31st Oct 2025

Dear Dr. Liu,

Thank you for submitting your revised manuscript "Telomere-to-Telomere Genomes of Diploids and Allotetraploids provide Insights into Peanut Genome Evolution and Improvement" (NG-A65542R4). It has now been seen by the original referees and their comments are below. The reviewers find that the paper has improved in revision, and therefore we'll be happy in principle to publish it in Nature Genetics, pending minor revisions to comply with our editorial and formatting guidelines.

Sincerely,

Wei Li, PhD
Senior Editor
Nature Genetics
www.nature.com/ng

Reviewer #4 (Remarks to the Author):

The authors have addressed all comments and suggestion of reviewers, thus, I suggest to accept it for publication..

To guide the scope of the revisions, the editors discuss the referee reports in detail within the team, including with the chief editor, with a view to identifying key priorities that should be addressed in revision and sometimes overruling referee requests that are deemed beyond the scope of the current study. In this case, we ask you to address the reviewers' comments in full. We expect you to address all the technical concerns raised by the Reviewers, especially those regarding the analysis of the estimated age of polyploidy (Reviewer #2) and the potential confounding of population structure for the GWAS results (Reviewer #1).

*1) Include a “Response to referees” document detailing, point-by-point, how you addressed each referee comment. If no action was taken to address a point, you must provide a compelling argument. This response will be sent back to the referees along with the revised manuscript.

*2) If you have not done so already please begin to revise your manuscript so that it conforms to our Article format instructions, available here.
Refer also to any guidelines provided in this letter.

(C) S83. The structural variant types include INS (insertion), DEL (deletion), CNV (copy number variation), INV (inversion), and TRANS (translocation).

(A)

(B)

Response: Thank you for your valuable comment. To identify candidate genes, we applied a stepwise bioinformatics approach. First, we filtered variants from the population dataset to retain high-quality genotypes. Using GAPIT software, we identified signal intervals by integrating these genotypes with phenotypic data. Homologous comparisons were conducted with known functional genes in model crops such as rice and *Arabidopsis*; candidate genes were confirmed using phenotype data. To refine the selection, we focused on candidate nut phenotype genes within 200 kb of the peak regions and prioritized those with annotated functions related to the observed phenotypes. Gene haplotype analysis was performed using population data, and the relationship between haplotypes and phenotypes was used to finalize candidate genes. Functional validation was carried out in *Brassica napus* and *Arabidopsis*.

- WGD analyses. Please use syntenic analysis, not only Ks, which can be sensitive to underlying demographic differences, eg

Response: Thank you for the suggestion. We have incorporated syntenic analysis alongside *Ks* to address potential demographic sensitivities and provide a more comprehensive assessment. These revisions are detailed in **Figure 6 (Supplementary Figure 4)** and **lines 192-196**.

GWAS outcomes (as shown in **Figure 5**), we evaluated PCA results and progressively excluded samples with confounding factors. We then extracted genotypes from the remaining samples and used peak signal intensity to evaluate the accuracy of the results, which allowed us to identify optimal signal loci.

Analysis of PCA-confounding samples showed that they belong to an admixed population, consistent with findings from the phylogenetic tree and population structure analysis, which indicate a high degree of admixture. The materials include cultivated varieties that have undergone improvement through the introduction of wild germplasm (e.g., yuania9102). Due to their significant proportion of cultivated ancestry, these samples continue to cluster with cultivated types.

- Where is the TreeMix analysis? Where are the *f* stats in tabular form? What were the significance values used. Etc.

Response: Thank you for your comments. We apologize for the oversight in failing to update the method description after TreeMix-related analyses had been removed from the main text and supplementary materials. In the revised text, we have removed all TreeMix-related content. Gene flow analysis between groups was conducted by calculating the *fd* statistic in sliding windows of 100 kb with a 10 kb step. Windows with fewer than three informative SNPs were excluded, as were windows with negative Patterson's (*D*) statistic values (related to *fd*) and those with $fd > 1$. The top 1% of introgressed regions were retained for subsequent gene selection and enrichment analyses. The *fd* statistic results have been added to **Supplementary Table 24**.

of the genes have GO annotations, whereas 50.87% have KEGG annotations (**Table 2 (Supplemental Table 7)**).

Thank you again for your valuable suggestions. Your insights have been instrumental in assuring the quality of the genome annotation and ensuring the reliability of subsequent analysis results.

G. References: appropriate credit to previous work?

Yes, in the introduction, M&M and results. The discussion lacks references.

Response: Thank you very much for your suggestions. In the revised text, we have updated the discussion section based on your recommendations and incorporated additional references.

H. Clarity and context: lucidity of abstract/summary, appropriateness of abstract, introduction and conclusions

The manuscript is clear and easy to read, although the discussion should be improved. The abstract summarizes well the results presented, as well as the introduction and the conclusions.

Reviewers' Comments:

Reviewer #1 (Remarks to the Author):

After enormous revisions, many of which entailed significant new bioinformatic research, this manuscript is now an excellent example of a paper with the interest level and attention to detail requisite for a journal such as Nature Genetics. I am satisfied with how the investigators followed up on my suggestions, and am also particularly impressed at how independent approaches (fastsimcoal being one of them) yielded similar dates. This consistency speaks to both cares taken to analyses as well as, of course, the appropriateness of the data to address the questions at hand. My only final suggestion - which I think is very important these days - is to revise all main-text figures such that they use similar (and best, color-blind compatible) color palettes. For example, Figs. 1 and 2 stand out very starkly in "standard" red, blue, and (bright) green colors, while Figs. 3 and 4, e.g., have much more muted colors that are easier to view.

Response:

We appreciate your summary of manuscript and encouraging comment.

Regarding your final suggestion about figure color palettes, we fully agree with your point about the importance of using color-blind friendly palettes. In the current revised manuscript, we have ensured that all figures adopt consistent and more accessible color schemes, with careful attention to color-blind compatibility. We have updated Fig. 1 and Fig. 2 to replace the bright red, blue, and green with more muted, color-blind friendly alternatives, in line with the colors used in Figs. 3 and 4.

We sincerely thanks for your thoughtful feedback, which has significantly improved the presentation of our work.

Therefore, out of 28,622 gene pairs, we would expect $28,622 \times 0.88 = 25,223$ to have $K_s=0$. (That number is pretty close to the B vs B comparisons from the paper's data which in one example is ~24,500 gene pairs having $K_s=0$)

Therefore, the genes with $K_s>0$ are unusually highly mutated. Filtering $K_s=0$ numbers is like trying to determine the average speed of cars on a highway by only measuring the fastest ones - you'd ignore the slower majority, leading to a distorted conclusion. The absence of genes with $K_s=0$ removes the necessary baseline for accurately estimating divergence on a short timescale, as the molecular clock relies on the full distribution of substitutions, including those that have not accumulated any. Having said this, the best way to date close divergences like this is SNP rates from, whole genome alignments.

Response:

Thank you for your detailed and insightful feedback on our work. I greatly appreciate your careful review of the figures and your explanations regarding the flaws in our plotting methods. The issues you raised, such as the incorrect labeling of the vertical axis, the inappropriate use of line charts to display frequency distributions, and the misrepresentation of the blue line, have been duly noted, and we agree that these elements need to be corrected to align with standard scientific data presentation. I also recognize that the K_s value is not suitable for closely related genomes, and I appreciate your suggestion to use SNP data from whole-genome alignments to better estimate the level of genomic divergence. We have removed the K_s -based results in the revised version in **lines 191-199&791-803**.

I apologize for the errors with the line chart color labeling and the misunderstanding between the last response version and the original publication. We regret any confusion this may have caused.

Additionally, as per your suggestion, we have calculated the time of polyploidization in the peanut genome using SNP data. By analyzing the SNP divergence between the A subgenome (aligned with V14167) and the B subgenome (aligned with K30076), we have estimated the polyploidization time.

SNP identification was performed using MUMmer. The modal divergence was statistically analyzed with a 10 kb window and a 1 kb overlap. The results indicated that the A subgenome had an average divergence of 21.5 (ranging from 19 to 23), while the B subgenome exhibited a significantly lower average divergence of only 1.25 (ranging from 1 to 2). Furthermore, the modal SNP frequency was also calculated, with a modal SNP frequency of 60.02 per 10,000 bases (ranging from 59.57 to 60.54) in the At subgenome, and a modal SNP frequency of 4.01 per 10,000 bases (ranging from 3.48 to 4.68) in the Bt subgenome (**Table 3 (Supplemental Table 10)**).

Further analysis suggests that the AA-At divergence occurred around 190,000 years ago, while the BB-Bt divergence took place approximately 10,000 years ago (**Table 3 (Supplemental Table 10)**). These results indicate that the whole-genome duplication event occurred approximately 10,000 years ago, which is consistent with previous estimates⁸.

These updated results have been incorporated into the revised version, specifically in **lines 200-209&787-790**. We believe this provides a more accurate and meaningful analysis of the divergence

time scale.

Once again, thank you for your valuable suggestions and thorough guidance, which have greatly improved the quality and accuracy of our research.

Table 3. The SNP rates in whole genome alignments.

Ref query	All snps	The modal divergence (10KB)	Genome size bp	SNPs/10KB	years
V14167_S245A	7148685	22	1200000000	59.57	187466
V14167_HN873A	7265256	22	1200000000	60.54	190522
V14167_HN51A	7220666	19	1200000000	60.17	189353
V14167_S83A	7177163	23	1200000000	59.81	188212
K30076_S245B	701635	2	1500000000	4.68	14720
K30076_HN873B	607475	1	1500000000	4.05	12744
K30076_HN51B	522102	1	1500000000	3.48	10953
K30076_S83B	575081	1	1500000000	3.83	12065

Why does Supplementary Figure 14 (B) use two different methods for the divergence dates Ks for the origin and coalescent-based genetic simulation for the others? The latter is giving dates I would expect (although they are stated with absurd precision (eg 9,491 years)).

Response:

Thank you very much for your insightful feedback. We initially intended to integrate the origin and divergence times of peanuts, which led us to combine both aspects. However, thanks to your guidance, we now understand that the KS results are not suitable for estimating divergence times in this context. As a result, in the new version, we have made corrections and retained only the results related to population divergence (**Figure 5(Supplementary Figure 14B)**). We sincerely appreciate your valuable suggestions.

Figure 5. Demographic history of peanut populations.

Reviewer #3 (Remarks to the Author):

Overall evaluation on the revised version

The authors have successfully resolved the concerns and questions risen during the first revision. The manuscript, although it contains a huge amount of information, is clear and easy to read. The results are well supported by several data sources (phenome2genome associations, transcriptomics...). As I mentioned in my first revision, the manuscript is well built, has interest and present relevant results not only for the peanut scientific community but also for the genomic community working in genome evolution and polyploidy.

Revised by Aureliano Bombarely on December 23rd, 2024

Response:

Thank you for your positive feedback and for recognizing the revisions made in response to the first review. We are pleased that the manuscript is clear and well-supported by various data sources. As you mentioned, we are glad that the manuscript's structure and findings are relevant not only to the peanut scientific community but also to the broader genomic community focused on genome evolution and polyploidy. Your feedback has been invaluable in improving the manuscript.

Once again, thank you for your thoughtful comments and support.

References

1. Zhang, Y. C. *et al.* Overexpression of microRNA OsmiR397 improves rice yield by increasing grain size and promoting panicle branching. *Nat Biotechnol* 31, 848–852 (2013).
2. Cho, S. H. *et al.* The rice narrow leaf2 and narrow leaf3 loci encode WUSCHEL-related homeobox 3A (OsWOX3A) and function in leaf, spikelet, tiller and lateral root development. *New Phytologist* 198, 1071–1084 (2013).
3. Dai, M., Hu, Y., Zhao, Y., Liu, H. & Zhou, D. X. A WUSCHEL-LIKE HOMEODOMAIN gene represses a YABBY gene expression required for rice leaf development. *Plant Physiol* 144, 380–390 (2007).
4. Otyama, P. I. *et al.* Genotypic characterization of the U.S. Peanut core collection. *G3: Genes, Genomes, Genetics* 10, 4013–4026 (2020).
5. Jung, S. *et al.* The High Oleate Trait in the Cultivated Peanut [*Arachis hypogaea* L.]. I. Isolation and Characterization of Two Genes Encoding Microsomal Oleoyl-PC Desaturases.
6. Chu, Y., Holbrook, C. C. & Ozias-Akins, P. Two alleles of ahFAD2B control the high oleic acid trait in cultivated peanut. *Crop Sci* 49, 2029–2036 (2009).
7. Yang, H. *et al.* Fine mapping of qAHPS07 and functional studies of AhRUVBL2 controlling pod size in peanut (*Arachis hypogaea* L.). *Plant Biotechnol J* 21, 1785–1798 (2023).
8. Bertoli, D. J. *et al.* The genome sequences of *Arachis duranensis* and *Arachis ipaensis*, the diploid ancestors of cultivated peanut. *Nat Genet* 48, 438–446 (2016).
9. Kochert, G. *et al.* RFLP AND CYTOGENETIC EVIDENCE ON THE ORIGIN AND EVOLUTION OF ALLOTETRAPLOID DOMESTICATED peanut, *Arachis hypogaea* (Leguminosae). *Am J Bot* 83, 1282–1291 (1996).
10. Moretzsohn, M. C. *et al.* A study of the relationships of cultivated peanut (*Arachis hypogaea*) and its most closely related wild species using intron sequences and microsatellite markers. *Ann Bot* 111, 113–126 (2013).
11. Seijo, G. *et al.* Genomic relationships between the cultivated peanut (*Arachis hypogaea*, Leguminosae) and its close relatives revealed by double GISH. *Am J Bot* 94, 1963–1971 (2007).
12. Robledo, G. & Seijo, G. Species relationships among the wild B genome of *Arachis* species (section *Arachis*) based on FISH mapping of rDNA loci and heterochromatin detection: A new proposal for genome arrangement. *Theoretical and Applied Genetics* 121, 1033–1046 (2010).
13. Grabile, M., Chalup, L., Robledo, G. & Seijo, G. Genetic and geographic origin of domesticated peanut as evidenced by 5S rDNA and chloroplast DNA sequences. *Plant Systematics and Evolution* 298, 1151–1165 (2012).
14. Leal-Bertioli, S. C. M. *et al.* Phenotypic effects of allotetraploidization of wild *arachis* and their implications for peanut domestication. *Am J Bot* 104, 379–388 (2017).
15. Krapovickas, A. & Gregory, W. C. Taxonomy of the Genus *Arachis* (Leguminosae). *Bonplandia* 16 (SUPL.), 1–205 (2007).
16. Freitas, F., Francisco, J. & Valls, M. *Genetic Variability of Brazilian Indian Landraces of Arachis Hypogaea* L. <https://www.researchgate.net/publication/5798151>.
17. Dillehay, T. D., Rossen, J., Andres, T. C. & Williams, D. E. Pre-ceramic adoption of peanut, squash, and cotton in Northern Peru. *Science* (1979) 316, 1890–1893 (2007).
18. Simpson, E. *et al.* *History Of Arachis Including Evidence Of A. Hypogaea* L. Progenitors History and Origin of the Genus.

Authors' responses to Reviewers Comments

We would like to express our sincere thanks for the insightful suggestions provided by the editor and the Reviewers, which have significantly contributed to the improvement of our manuscript. The following are our responses to each of the Reviewers' comments (in blue colour).

Reviewer #1 (Remarks to the Author):

I am entirely satisfied with the authors' acceptance of my suggestion re: colorblind palettes: "Regarding your final suggestion about figure color palettes, we fully agree with your point about the importance of using color-blind friendly palettes. In the current revised manuscript, we have ensured that all figures adopt consistent and more accessible color schemes, with careful attention to color-blind compatibility. We have updated Fig. 1 and Fig. 2 to replace the bright red, blue, and green with more muted, color-blind friendly alternatives, in line with the colors used in Figs. 3 and 4."

Regarding their responses to the other reviewers, in a short look, they seem well-considered, but I'm afraid I don't have time to carefully evaluate them.

Authors' response: We are thankful to the Reviewer # 1 for appreciating the revised version of the MS.

Reviewer #2 (Remarks to the Author):

Please see uploaded file. It contains formatting of text and graphs that can't be entered into this box. If there are any issues with the uploaded file, please contact me so that we can address the problems.

Authors' response: We are thankful to the Reviewer # 2. Detailed response to other points is provided in the following sections.

Reviewer #3 (Remarks to the Author):

The author did resolve most of my concerns and questions in the previous revision. I think that the new version of the manuscript has been improved after a new round of interactions with other reviewers. From my point of view the manuscript it is good to

go.

Authors' response: Sincere thanks to the Reviewer # 3 for appreciating the revision.

Reviewer2' Comments:

Review of: “Telomere-to-telomere genomes of diploids and allotetraploids provide insights into the origin, evolution, and domestication of peanuts”

Despite its volume, the paper offers no new insights

The multitude of tables, graphs, and verbose text only serves to obscure the absence of substantive new conclusions. With the advent of long-read sequencing, assembling large and complex genomes have become relatively routine. The paper is poorly referenced, represents previously known findings as new, misrepresents prior research, and contains numerous errors, unsupported statements, and wild speculations. It has undergone very substantial changes to its fundamental conclusions during review—this removes some of the more obvious mistakes—but how can the credibility of the work remain intact? Indeed, after looking further, I highlight below more major problems below:

The paper's current title doesn't reflect its content. A suitable title could be: “**Extreme similarity of peanut genomes confirms a recent single origin**”

Authors' response: Thank you very much for your suggestion. We have addressed suggestions of the Reviewer 2 while revising the MS. We have also updated the title to: '**Telomere-to-telomere genomes of diploids and allotetraploids provide insights into genome organization, evolution, and agronomic traits for peanut improvement.**'

Origin

There are no new insights into the origin of *Arachis hypogaea* in this paper. In reply to my question what are the insights, the authors replied in three points:

- 1) “we refine the timeline of tetraploid formation to approximately 10,000 years ago, consistent with earlier studies” – this was previously reported by Bertoli et al 2016.
- 2) “our centromere length analysis offers new insights into structural differences between subgenomes.” – this isn’t an insight about the origin, it’s about the structural differences of the component subgenomes.
- 3) “The analysis suggests that a diploid B species may have diverged from a diploid A species before hybridizing with an A genome to form the tetraploid.” - The divergence of the A and B subgenomes is well known, as is their union into the allotetraploid. The principle of divergence is obvious, it’s implied by the concept of evolution. The allotetraploid nature of peanut via the union of A and B subgenomes was first reported by Husted in 1936. The species origin of the A and B subgenomes has been the subject of extensive research that is now summarized in the new version of the paper in lines 62-88. The dates of divergence of the A and B subgenomes have been previously estimated as 3.5 Mya (Nielen et al 2011), then refined to 2.3–2.9 Mya (Moretzsohn et al 2012), then further refined to ~2.16 Mya by Bertoli et al 2016).

There are no new insights about the origin of *Arachis hypogaea* in this paper.

Authors’ response: Thank you very much for your suggestion.

1) We agree with Reviewer 2’s observation that the time of the first tetraploid formation, approximately 10,000 years ago, was reported in the article by Bertoli et al. (2016) ¹. However, subsequent studies have shown discrepancies regarding the formation time of the tetraploid. For instance, Chen et al. (2019) assembled the Fuhuasheng genome and compared it with previously published wild genomes, suggesting that the tetraploid formation time was less than 0.18 million years ago². Yin et al. (2020) analyzed the wild tetraploid Monticolar and wild materials published in 2019, estimating the tetraploid formation time of Monticolar to be around 11,690 years ago³. Additionally, in 2019 and 2020, Zhuang et al. and Bertoli et al. conducted assemblies of Shitouqi and Tifrunner, respectively. Based on the divergence relationship between tetraploid subgenomes and diploid genomes, they estimated significantly different times for the tetraploid divergence⁴⁻⁶. Zhuang et al. (2019 & 2020) estimated the tetraploid formation time to be around 45 million years ago, whereas Bertoli et al. (2016 & 2020) suggested

a time of approximately 10,000 years ago. Meanwhile, Zheng et al. (2024) re-sequenced the chloroplast and whole genome of peanuts and inferred the tetraploid divergence time. They proposed that the differences in tetraploid formation time between Ahh (<10,000 years ago) and Ahf (0.42-0.47 million years ago) might be attributed to the fact that the reference genomes come from different subspecies⁷. Thus, the time of tetraploid formation remains a subject of ongoing debate.

Additionally, due to the limitations of sequencing and assembly technologies, many published diploid and tetraploid genomes contain numerous contigs and gaps, which significantly affect the accurate identification of SNP variations. Moreover, since peanut has two subspecies and six varieties, predicting the tetraploid formation time based on a single subspecies or variety is inaccurate. We should mention that we should not just remain focused only studies from one group (Moretzsohn et al 2012; Bertioli et al 2016). We should consider all other studies that have also been published in top class journals such as Nature Genetics, Molecular Plant, etc.

In the present study, we constructed six T2T genomes, including two wild diploids and four varieties. This approach not only allows for accurate identification of variations between the tetraploid subgenomes and their diploid progenitors, but also enables the analysis of whether differences among subspecies/varieties affect the prediction of tetraploid formation time. Therefore, we believe that our analysis is crucial and necessary for accurately determining the tetraploid formation time. This result represents the application of complete genomes for variant identification and divergence time analysis in peanuts.

2) Compared to the centromere length in the diploid genome, there are significant differences in the centromere length and structure of the tetraploid B subgenome. As no complete centromeres of diploids and allotetraploids were reported and analyzed, our study provides insights of centromere evolution, enabling the formation and stable inheritance of the tetraploid.

3) As the Reviewer mentioned, there is general agreement in different studies regarding the divergence time of the wild diploid AB genome, which is approximately around 2 million years ago. However, the **origin relationship** between the A and B genomes (A to B or B to A) has been rarely reported. Using centromere repeat units for phylogenetic tree analysis, we have preliminarily identified that the diploid B species **may have originated from** a diploid A species. This result represents the first attempt to explore the divergence relationship of the diploid subgenomes (A and B) using centromere repeat units.

Building upon previous research, we have conducted a systematic analysis of the origin of multiple peanut varieties using multiple T2T genomes. In particular, we have further clarified the tetraploid formation time and analyzed the divergence relationship of the diploid genomes through centromere structural variation. This study holds significant importance for understanding the origin of peanuts.

Therefore, we don't think that it is a fair assessment of the MS and mentioning that "There are no new insights into the origin of *Arachis hypogaea* in this paper".

Evolution

There are no new insights into the evolution of *Arachis hypogaea* in this paper. In reply to the question as to what are the insights, the authors replied:

"Gypsy-type LTR retrotransposons showed distinct insertion times between the At and Bt subgenomes..." - Distinct insertion times of component genomes are to be expected with a divergence of more than 2 Mya, and have been reported before. Nielen et al (2010) reported at least threefold higher copy number of the most abundant Ty3-gypsy retrotransposon in the A-genome ancestor compared to the B-genome ancestor. Bertioli et al (2013) reported of the ten most abundant LTR retrotransposons that "Almost all dateable transposition events occurred <3.5 million years ago, the estimated date of the divergence of A and B genomes." and "The activity of these retrotransposons has been a very significant driver of genome evolution since the evolutionary divergence of the A and B genomes."

How insightful can the paper's evolutionary analysis be, if it failed to alert the authors that the original manuscript version had overestimated the origin of peanut by more than a quarter of a million years?

Authors' response: Thank you very much for your suggestion.

We carefully reviewed the articles mentioned by the Reviewer and summarized them as follows: Nielen et al. (2010) focused on the FIDEL-aretrovirus-like retrotransposon in the AB subgenomes and emphasized the role of FIDEL in the evolution and divergence of different *Arachis* genomes⁸. Bertoli et al. (2013) utilized 27 *A. duranensis* BAC clones as probes and employed the BAC-FISH method. They found that the sequences of 14 BAC clones revealed complete and truncated copies of ten abundant long terminal repeat (LTR) retrotransposons. These ten LTR transposition events occurred less than 3.5 million years ago and have been a significant driver of genome evolution since the evolutionary divergence of the A and B genomes⁹.

Due to limitations in sequencing and assembly technologies, the studies by Nielen et al. (2010) and Bertoli et al. (2013) could not provide a comprehensive genome-wide analysis of various transposon types and insertion times, nor could they explore the roles of different transposons in genome evolution. Additionally, due to material constraints, these studies were unable to perform identification across multiple varieties or subspecies.

In contrast, our study, based on the polyploid formation timeline, subdivides different types of transposable elements (TEs) based on their insertion times. We identified insertion events that may have occurred before polyploid formation and compared the changes in TEs before and after tetraploid formation. Our findings showed that multiple TEs present in the diploid genomes were absent from the tetraploid subgenomes, suggesting complex evolutionary dynamics and potential differential retention or loss of TEs during polyploidization. These results, which have not been previously reported, highlight the importance of the T2T genome assembly for further research on TE dynamics in peanut evolution.

We believe that our study, based on T2T genomes, provides a better understanding of the expansion of LTR retrotransposons in the AB subgenomes and will significantly contribute to advancing research on peanuts. Moreover, our findings offer valuable insights that could be applied to the study of TEs in other species as well. As mentioned earlier, if our studies provide different/ better results than reported by Bertoli et al. doesn't mean that our study is useless and doesn't provide new insights.

Domestication

There are no new insights into the domestication of *Arachis hypogaea* in this paper. In reply to the question as to what are the insights, the authors replied:

“The total number of SVs was significantly higher in the comparison of At_vs_V14167 (average: 70,439) than in Bt_vs_K30076 (average: 14,917)...”, their analysis then builds off this. - A greater number of SVs is completely expected between At_vs_V14167 than Bt_vs_K30076, because *A. duranensis* V14167 is more distantly related to the A subgenome donor than *A. ipaënsis* is to the B subgenome ancestor.

Closer representatives of the A subgenome donor, *A. duranensis* from Rio Seco, have been identified. Why didn't the authors sequence one of these?

How insightful can the paper's analysis of domestication be, given that it failed to alert the authors that their inference of “multiple origins of the B subgenome” in the original version of their manuscript was incorrect?

Authors' response: Thank you very much for your suggestion.

In terms of domestication, our analysis focused on comparing structural variations (SVs) between subgenomes, which included comparisons not only with wild diploids but also between different tetraploid varieties. In addition to using the wild diploid as the reference, we also used the cultivated tetraploid S83 as the reference to identify variations between tetraploid varieties. These varieties, during the domestication process, exhibit diverse phenotypes, such as increased pod number, seed size, oil content, and plant architecture. These phenotypic differences are influenced by various selective pressures. The primary cause of these phenotypic variations can be traced back

to genotype differences. The construction of the T2T genome not only enables the accurate identification of large structural variation (SV) differences but also allows for the discovery of genes related to these variations, followed by functional validation.

In this study, as per Reviewer 1's suggestion, we categorized the different SVs and conducted separate enrichment analyses, which will be crucial for the subsequent identification of key genes in peanut domestication. This will provide valuable insights into the genetic underpinnings of important traits that have been selected for during the domestication of peanuts.

Based on the SV analysis, we identified a 13.23 Mb-long insertion on Chr14, and visualized its coverage using both HiFi and Illumina reads. The accuracy of this structural variation (SV) was further confirmed through Hi-C heatmap analysis. Statistical analysis revealed that all samples in the G5 population contained this SV, while 23% and 15% of samples in the G4 and G3 populations contained it, respectively. No samples in the G2 population contained this SV (Supplementary Figures 10A-10E). These results suggested that this SV could be associated with the differentiation between the G5 and G4 populations, which both belong to the var. *hypogaea* subspecies. In this SV region, we identified 53 homologous genes with over 70% sequence identity and an e-value of less than $1e-10$, including several rice functional genes related to plant architecture and organ development, such as OsA7, OsWOX3A, OsWOX3B, OsIAA17, OsLAC, and OsRLCK107. Among these, two genes, chr14.1804 and chr14.1807, were found to be tissue-specific, particularly expressed in roots, stems, and leaves but not in cotyledons, shells, seeds, or pegs (Supplementary Fig. 12). These two genes are homologous to OsLAC, which is a target gene of OsmiR397. Studies suggest that OsLAC regulates various developmental processes in rice by modulating the brassinosteroid (BR) signaling pathway. Overexpression of OsLAC leads to shortened stems, upright leaves, and reduced grain size, phenotypes resembling those seen in BR-deficient plants. Moreover, OsmiR397, which negatively regulates OsLAC expression, enhances BR signaling and alters plant architecture by increasing leaf angles. These

findings indicate that OsLAC, through its interaction with the BR pathway, plays a vital role in regulating plant growth and development, offering potential strategies for improving crop architecture and yield through the manipulation of hormone signaling pathways¹⁰.

Another gene, chr14.1829, which is homologous to the WOX3A/B gene. Studies indicate that both OsWOX3A and OsWOX3B genes play crucial roles in the organ development of rice, especially in the development of leaves, tillers, lateral roots, and lateral organs. In the case of the OsWOX3A mutant, the plant exhibits narrow and curled leaves, an increase in tiller number, a decrease in lateral roots, and narrower grains. This phenotype is associated with restricted lateral growth of the leaves, fewer longitudinal veins, and the enlargement of bubble-like cells. Moreover, OsWOX3A also plays a key role in the development of the outer glume (lemma), with mutations leading to abnormal glume morphology, affecting the seed shape¹⁰. At the molecular level, OsWOX3A is primarily expressed in the vascular tissues of rice, and mutations lead to changes in the expression of genes related to leaf development, hormone transport, and other pathways, resembling the phenotypes seen in PIN gene mutants.

On the other hand, OsWOX3B plays an essential role in the symmetry of leaf development. The OsWOX3B mutant displays asymmetrical leaves and sometimes forked leaf shapes, which leads to abnormal development of lateral organs, affecting leaf width and morphology. Mutants also show an increase in tillers, longer panicles, and a greater number of grains per panicle. Furthermore, the vascular bundles in the flag leaf (the leaf above the panicle) increase in number, and the spacing between small vascular bundles widens, further affecting the growth and development of rice¹¹.

Otyama et al. (2020) classified 787 accessions from the U.S. Peanut core collection and 12 commercial varieties into Clades 1 to 4 using the *Arachis*_Axiom2 SNP array data¹². Phenotypic observations showed that in Clade 1, "most accessions are classified as the hypogaea botanical variety and exhibit a 'mixed' or hypogaea pod shape (60.0%, 44.4%, 40.0%). The growth type varies widely, with an even distribution between erect, bunch,

spreading-bunch, mixed, and prostrate types."

Further analysis reveals that Clade 1 in our study corresponds to the G4+G5 populations. Based on phylogenetic relationships and material information, we found that both G4 and G5 belong to var. hypogaea. Field surveys indicate that the G5 population has a larger tillering angle (with a higher proportion of prostrate types), whereas the G4 population has a larger proportion of erect types. This is consistent with the observed variation in growth types within Clade 1.

Combining the functional annotation and expression results of the 109 SV-genes, we hypothesize that the growth type difference between the G4 and G5 populations may be linked to the presence of this structural variant (SV).

Therefore, we believe that comparing the structural variations (SVs) between the diploid genome and different tetraploid subgenomes is of great significance for exploring genomic changes during the tetraploid domestication process. At the same time, SV analysis between different tetraploid varieties plays a crucial role in identifying key domestication candidate genes that have been subject to selection. This comparative approach not only helps to elucidate the genomic alterations that have occurred during the domestication of peanuts but also aids in pinpointing specific genetic changes that have contributed to the selection of important traits. Identifying these SVs and their associated genes will provide valuable insights into the underlying mechanisms of domestication and could inform future breeding strategies for enhancing peanut crops.

Above explanations indicate and prove our point about new insights on domestication.

The assemblies are incomplete

A key feature of the paper is that it reports "telomere to telomere assemblies".

In fact, four half-chromosome arms are missing:

Missing region 1)

More than 6 Mb (about half the euchromatic chromosome arm, where most of the chromosome's genes are) is missing from chr05 of HN873. The sequence is indicated by the red arrow in the below plot of HN873 vs the Tifrunner reference genome.

Missing region 2)

About half the euchromatic chromosome arm is missing from chr15 of HN873. The sequence is indicated by the red arrow in the below plot.

Missing region 3)

About half the euchromatic chromosome arm is missing from chr05 of HN51. This is indicated by the red arrow in the below plot.

Missing region 4)

About half the euchromatic chromosome arm is missing from chr15 of HN51. This is indicated by the red arrow in the below plot of HN51.

The terminal regions of all these four chromosomes are also completely lacking characteristic telomeric sequences. The explicit claim made in the paper that there are 40 telomere sequences in the four sequenced tetraploid genomes is simply incorrect. As is their gross misrepresentation that the previously sequenced genomes: Tifrunner, Shitouqui, Fuhuasheng, *A. duranensis* and *A. ipaënsis* have 0 telomeric sequences! (see

eg. Table 1) – Many of the chromosome ends of previous assemblies do, in fact, have the telomeric sequences.

For comparison here is a plot of two complete chromosomes.

Note how the line in the plot runs from corner to corner, indicated by blue arrows:

A major selling point of the paper is its analysis of genomic differences—but how reliable and informative could that analysis have been since the absence of four half-chromosome arms went unnoticed?

The paper misrepresents genome similarities

These plots reveal another major shortcoming of the paper: it gives a misleading impression of extensive, complex differences between the genomes. In reality, the genomes are strikingly similar to current reference genomes. The lines in the plots above are composed of dots indicating 4,000 bp windows of identical sequence. Aside from the centromeric regions, which are highly repetitive and gene-poor, and a few deletions and reciprocal exchanges, the genome sequences are almost identical to the existing diploid and tetraploid reference sequences. The authors have presumably “talked-up” the differences to make their hypothesis-free research seem more significant.

Authors' response: We greatly appreciate your thorough analysis.

Upon receiving this feedback, we immediately verified the data and discovered that the

issue arose because the Reviewer had used the **V1-gapfree version** of the genome uploaded on May 30, 2024. However, prior to our first revision (2024.12.08, NG-A65542R), we had updated our assemblies and the **latest T2T genomes** were provided in the Data Availability section (lines 1103-1108). It seems that the Reviewer used the older version of the assembly and made his/her remarks.

Please note that all related data analyses, present in this MS (version) is based on the update version of genome assemblies. However, we wonder, that if the Reviewer has not used the correct version of genome assemblies (probably he/she just wants to kill the paper, and without even going through the update genome assemblies, he/she just made negative comments on the basis of previous version.

When we compared our T2T genomes with the Tifrunner.gnm2 genome used by the Reviewer, we observed a high degree of collinearity between these chromosomes. Below are the details of the genome update process and the associated synteny analysis results, including data processing and alignment information:

Telomere Deletion Validation:

Based on the Reviewer's alignment results, we verified the genomic collinearity and discovered that the Reviewer used the **gapfree-V1** version of the genome from May 30, 2024. This version has incomplete telomere sequences, which resulted in the telomere deletions mentioned by the reviewer (**shown in the left image in the Figure below**). The middle image shows our verification results using the gapfree-V1 version. And the right image shows the alignment results from the **T2T-V2** version with the same genome used by the Reviewer (arahy.Tifrunner.gnm2.J5K5.genome_main.fna), and it showed higher consistency, with no fragment deletions present.

Collinearity Results of Different Genome Versions with the Tifrunner.gnm2 Genome. gapfree-V1: Left and Middle Figures, T2T-V2: Right Figure, Chromosome start: Red and Blue arrows indicate.

1) HN873.chr05-VS-Tifrunner.gnm2.chr05

The image on the right shows the latest version of the genome, with no deletions observed.

2) HN873.chr15-VS-Tifrunner.gnm2.chr15

The image on the right shows the latest version of the genome, with no deletions observed.

3) HN51.chr05-VS-Tifrunner.gnm2.chr05

The image on the right shows the latest version of the genome, with no deletions observed.

4) HN51.chr15-VS-Tifrunner.gnm2.chr15

The image on the right shows the latest version of the genome, with no deletions observed.

5) The collinearity of different accessions on chr05 and chr15 shows that the V2 version of the genome exhibits a high level of consistency in chromosomal collinearity, with no telomere deletions (regions in red boxes).

Additionally, we performed a synteny analysis using the published Tifrunner.gnm2 genome as the reference, comparing it with our assembled 6 T2T genomes and 6 published genomes (the K30076 (public) genome from Bertoli et al. (2016)¹; Adu (public), Amon, mH8, and NHD108 genomes from Zhao et al. (2025)¹³; YZ9102 from Wang et al. (2025)¹⁴. Except for YZ9102, which is a T2T genome, the contig

sequences in the other five genomes were deleted prior to alignment, leaving only chromosome sequences.

The results showed a high degree of synteny with Tifrunner, with no evidence of telomere deletions. This result is also added in **Supplementary Figure 4**. The results are as follows:

Based on the above results, we believe that the issue of telomere deletions mentioned was caused by the incorrect use of the gapfree-V1 version (2024.05.30) of the genome, and there is no issue with assembly completeness in the updated T2T-V2 genomes. The genomes used in the above analysis can also be downloaded from the following databases and links:

1. Figshare:

https://figshare.com/articles/dataset/T2T_genome_assembly_in_peanut/28883636

2. NGDC:

https://ngdc.cncb.ac.cn/gwh/search/advanced/result?search_category=&search_term=&source=0&query_box=PRJCA026588

3. Google Drive:

https://drive.google.com/drive/folders/1Y_HoDMkFaPXgOMXDKHs3SyWvUIhFAfZe

Once again, we appreciate the Reviewer's feedback, and we believe that viewing and analysis of the updated T2T genome can address the Reviewer's concerns.

Genome-wide association studies yielded no new verified associations

In response to my criticism that nearly all of the 4,638 loci reported as associated with agronomic traits lack verification, the authors cite just two FAD gene marker-trait associations-both of which are already well known and have been published in multiple prior studies.

The FAD gene sequences and their role in the high oleic trait were reported in the early 2000s by Jung et al (2000), López et al 2000 and López et al 2002 (see also Refs 35-37 in the paper).

The authors also give the case of *AhRUVBL2*, but without any details of verification, only saying “we have prioritized these representative traits for candidate gene mining, haplotype analysis, and experimental validation.”

Authors' response: Thank you very much for your suggestion.

Regarding the results of the GWAS analysis, we considered the points exceeding the threshold as candidate loci and performed statistical analysis, a method commonly used in several studies. By combining the 200kb linkage interval around the GWAS Manhattan peak points, we selected previously functionally validated genes, *FAD*¹⁵⁻¹⁷ and *AhRUVBL2*¹⁸, as positive controls to confirm the accuracy of our GWAS results. Furthermore, based on the population haplotype analysis, we identified a significant correlation between the variation in the *FAD* gene and the high-oil and low-oil populations, which is consistent with previous studies.

For the *AhRUVBL2* gene, we observed a clear signal in the GWAS Manhattan results, and it lies within the 200kb linkage interval around the peak points. Previous studies have shown that this gene is a key regulator of seed size. We used these two genes as

positive controls to demonstrate the accuracy of our GWAS results.

Regarding the Reviewer 2's comment about "prioritizing these representative traits for candidate gene mining, haplotype analysis, and experimental validation," we selected major effect genes that influence oil content, a key trait for peanut quality, and genes that regulate seed size, which is critical for yield and development, for a series of validation work. The results of these validations are presented in the main text.

It is also important to mention that GWAS analysis provides large number of genetic loci/genes associated with various agronomic traits, and verification/ validation can be done only for few genes as examples. Therefore, such a criticism that validation has been done only for two genes and that too for well-established genes is not justified.

The comments above highlight some of the most important problems with the paper. Below is a selection of comments on the narrative from the Discussion

Representing previously known findings as new

Bian et al, the first results discussed in Discussion section lines 530-532 "Based on the analysis of TE insertion timing, the insertion of Gypsy elements was identified as the primary factor contributing to the larger size of the B genome/subgenome relative to the A genome/subgenome."

Compare to results from Bertoli et al 2016 "All *A. ipaensis* pseudomolecules were larger than their *A. duranensis* counterparts... This is partly because of a greater frequency of local duplications and higher transposon content in *A. ipaensis*." Furthermore, this is because of deletions after inversions (not just insertions) "Size differences between homeologous chromosomes that were differentiated by large rearrangements tended to be greater than those between collinear ones ($r(6) = 0.65$, $P < 0.05$; Supplementary Table 14). Because the *A. duranensis* chromosomes that have undergone inversions are smaller than expected, it is evident that, in this dynamic, on balance, the elimination of DNA has predominated over its accumulation."

This is gone over in more detail by Ren et al 2018.

Bian et al, Lines 554-555 “These findings suggest that a diploid B species diverged from a diploid A species and subsequently hybridized with an A genome to form a tetraploid.”

– this is, of course, completely established

Bian et al, Lines 562-563 “Based on the results of the phylogenetic tree and population structure, we inferred that the differentiation of cultivated peanuts into four major groups.” – this is, of course, completely established

Authors’ response: Thank you very much for your valuable suggestions. We have revised discussion sections based on your feedback.

Wild Speculation

Lines 535-537 “Significant TE insertion events were also observed in the A genome/subgenome during the S3 period (100,001–300,000 years ago), suggesting that these events created favorable genomic conditions for tetraploid formation.”

Lines 539-540 “This discrepancy may largely be attributed to differences in genome stability and the functional redundancy of TEs.”

Lines 548-549 “The difference in centromere length between cultivated peanuts and wild diploids related to chromosome pairing during meiosis.”

Authors’ response: Thank you very much for your valuable suggestions. Although we don’t agree with comments of the Reviewer, we have considered some suggestions and revised the MS accordingly.

Errors

Lines 568-570 “var. *fastigiata* types are primarily cultivated for consumption ... In contrast, var. *hypogaea* types are predominantly grown for oil extraction.” In fact, virtually the entire production of the USA, Argentina and Brazil is var. *hypogaea* types for direct consumption.

Lines 571-573 “Furthermore, the differentiation between G4 and G5 primarily arises from differences in planting methods. G4, with its upright growth habit, supports higher

planting densities, thereby increasing yield.” This is simply untrue; prostrate growth habitpeanuts have a longer growing season and, consequently, generally a much higher yield.

Their cultivation is generally preferred where the rainy season is long enough, especially where harvest is mechanized.

Authors’ response: Thank you very much for your valuable suggestions. We have revised and corrected this section based on your feedback.

References

1. Bertoli, D. J. *et al.* The genome sequences of *Arachis duranensis* and *Arachis ipaensis*, the diploid ancestors of cultivated peanut. *Nat Genet* 48, 438–446 (2016).
2. Chen, X. *et al.* Sequencing of Cultivated Peanut, *Arachis hypogaea*, Yields Insights into Genome Evolution and Oil Improvement. *Mol Plant* 12, 920–934 (2019).
3. Yin, D. *et al.* Comparison of *Arachis monticola* with Diploid and Cultivated Tetraploid Genomes Reveals Asymmetric Subgenome Evolution and Improvement of Peanut. *Advanced Science* 7, (2020).
4. Zhuang, W. *et al.* The genome of cultivated peanut provides insight into legume karyotypes, polyploid evolution and crop domestication. *Nat Genet* 51, 865–876 (2019).
5. Zhuang, W. *et al.* Reply to: Evaluating two different models of peanut’s origin. *Nature Genetics* vol. 52 560–563 Preprint at <https://doi.org/10.1038/s41588-020-0627-0> (2020).
6. Bertoli, D. J., Abernathy, B., Seijo, G., Clevenger, J. & Cannon, S. B. Evaluating two different models of peanut’s origin. *Nature Genetics* vol. 52 557–559 Preprint at <https://doi.org/10.1038/s41588-020-0626-1> (2020).
7. Zheng, Z. *et al.* Chloroplast and whole-genome sequencing shed light on the evolutionary history and phenotypic diversification of peanuts. *Nat Genet* 56, 1975–1984 (2024).
8. Nielen, S. *et al.* FIDEL-a retrovirus-like retrotransposon and its distinct evolutionary histories in the A- and B-genome components of cultivated peanut. *Chromosome Research* 18, 227–246 (2010).
9. Bertoli, D. J. *et al.* The repetitive component of the A genome of peanut (*Arachis hypogaea*) and its role in remodelling intergenic sequence space since its evolutionary divergence from the B

- genome. *Ann Bot* 112, 545–559 (2013).
10. Zhang, Y. C. *et al.* Overexpression of microRNA OsmiR397 improves rice yield by increasing grain size and promoting panicle branching. *Nat Biotechnol* 31, 848–852 (2013).
 11. Dai, M., Hu, Y., Zhao, Y., Liu, H. & Zhou, D. X. A WUSCHEL-LIKE HOMEODOMAIN gene represses a YABBY gene expression required for rice leaf development. *Plant Physiol* 144, 380–390 (2007).
 12. Otyama, P. I. *et al.* Genotypic characterization of the U.S. Peanut core collection. *G3: Genes, Genomes, Genetics* 10, 4013–4026 (2020).
 13. Zhao, K. *et al.* Pangenome analysis reveals structural variation associated with seed size and weight traits in peanut. *Nat Genet* (2025) doi:10.1038/s41588-025-02170-w.
 14. A telomere-to-telomere genome assembly of the cultivated peanut. doi:10.6084/m9.
 15. Jung, S. *et al.* *The High Oleate Trait in the Cultivated Peanut [Arachis Hypogaea L.]. I. Isolation and Characterization of Two Genes Encoding Microsomal Oleoyl-PC Desaturases.*
 16. Chu, Y., Holbrook, C. C. & Ozias-Akins, P. Two alleles of ahFAD2B control the high oleic acid trait in cultivated peanut. *Crop Sci* 49, 2029–2036 (2009).
 17. Nawade, B. *et al.* Insights into the Indian peanut genotypes for ahFAD2 gene polymorphism regulating its oleic and linoleic acid fluxes. *Front Plant Sci* 7, (2016).
 18. Yang, H. *et al.* Fine mapping of qAHPS07 and functional studies of AhRUVBL2 controlling pod size in peanut (*Arachis hypogaea* L.). *Plant Biotechnol J* 21, 1785–1798 (2023).

Authors' responses to Reviewers Comments

We sincerely thank the Editor and Reviewers for their insightful feedback, which has greatly contributed to the improvement of our manuscript. Below are our responses to each reviewer's comments (highlighted in blue).

Reviewer #1 (Remarks to the Author):

The authors, in their response, present a convincing case with associated graphics that appears to demonstrate that a reviewer mistakenly analyzed an earlier version of the assembly in question. I have no further comments for improvement.

Authors' response: We sincerely thank Reviewer #1 for your thoughtful feedback and for recognizing the revisions we made in the manuscript. We greatly appreciate the time and effort invested in reviewing our work.

Reviewer #3 (Remarks to the Author):

As I stated in previous revisions, I think that the authors resolved all the concerns that I rose during my first revision. Furthermore, I think that they did an excellent job answering and resolving not only my questions and concerns but also others risen by the other reviewers. I think that the manuscript has enough merit for its publication.

Authors' response: Thank you very much for your kind words and for recognizing the efforts we have made in revising the manuscript. We truly appreciate your constructive feedback, which has been invaluable in improving the quality of our work. We are grateful for your support and look forward to the manuscript's potential publication.

Reviewer #4 (Remarks to the Author):

Comparative Analysis & Novelty:

Previous studies have already addressed topics such as genome assembly, selective

sweeps, evolutionary history, and phenotypic diversification. The authors should represent the unique contributions through the differences between previous studies and this study.

Authors' response: We appreciate your insightful suggestions. In the revised version, we have added a comparison with previous studies on genome assembly, particularly emphasizing our novel contribution of assembling the T2T genomes of two wild diploid species and four representative cultivated varieties. A comparison with published genomes revealed that, aside from the previously published YZ9102 genome, which has already achieved a T2T level, the other genomes still contain numerous contig sequences. Importantly, no T2T-level genome for wild diploid species has been published to date, marking a significant advancement in accurately identifying the timing of tetraploid formation and tetraploid variations. For instance, we found chromosomal exchanges on chr07, chr08, chr17, and chr18 in the tetraploid, which were already present in the ancestral diploid, not occurring after the formation of the tetraploid.

Regarding the population analysis section, we have summarized key previous studies on peanut populations. Previous research primarily focused on the following:

1. **Improved Analysis from Landrace to Cultivated Varieties:** For example, Liu et al. (2022) performed a population analysis of 203 cultivated varieties. In their selective sweeps analysis, they used *Fst* and *XP-CLR* to examine landrace and cultivated varieties, identifying signals of peanut improvement.
2. **Comparison of Peanuts in Northern and Southern China:** Lu et al. (2024) analyzed a population of 390 peanut materials, focusing on selective elimination between northern and southern China, as well as non-Chinese materials, identifying relevant regions and genetic diversity.
3. **Phenotypic Differences between Two Subspecies:** Zheng et al. (2025) divided 353 tetraploid cultivated peanut materials into two subspecies, *Ahh* and *Ahf*, and analyzed candidate genes related to traits such as flowering patterns and seed inner tegument color.

In our study, based on the T2T genome, we have expanded upon previous work by identifying variations within a population of 521 representative accessions. For the first time, we compared genetic differentiation and introgression (gene flow) regions among the four distinct cultivated varieties, identifying significant asymmetric differentiation and introgression patterns. Additionally, in response to Reviewer 1's suggestion, we evaluated the impact of transposable elements (TEs) on the top 1% of these selection and introgression regions.

Once again, thank you for your valuable suggestions. These revisions have greatly contributed to the improvement of our manuscript.

Origin & Evolution Section:

The contents on Origin and Evolution should be toned down, as these aspects have been provided in previous studies. The authors should focus more on their novel findings.

Authors' response: We thank the Reviewer for this suggestion. In the revised version, we have streamlined and revised the sections on peanut origin and evolution to minimize overlap with previous studies, thereby placing greater emphasis on our novel findings.

SV-Domestication Relationship:

Authors should strengthen the relationship between structural variations (SVs) and domestication. The current manuscript does not sufficiently explain the functional relationships among genes, SVs, and domestication traits.

Authors' response: We appreciate your valuable suggestion. In addition to identifying genes related to plant architecture in the 13.2 Mb variation on chromosome 14, we have further identified and enriched four representative peanut-specific SV variations in the revised version, using the wild diploid as the reference (**lines 281-301**). We have also performed enrichment analyses of the specific genes from different variations, which allowed us to further elucidate the pivotal role of SVs in the domestication process.

Thank you once again for your input. These additions significantly enhance our understanding of peanut domestication.

Genome Similarities:

I agree with the authors' responses regarding genome similarities.

Authors' response: Thank you very much for your recognition. We appreciate your acknowledgment of our responses regarding genome similarities.

GWAS Analysis:

In addition to the known loci (FAD and AhRUVBL2), the authors are encouraged to include genetic mechanism analyses for one or two additional phenotypic traits to enhance the GWAS findings.

Authors' response: Thank you very much for your valuable suggestion. In the revised version, we have expanded our analysis to include two additional phenotypic traits: Kernel Dehydration Rate (KDR) and Arachidic Acid content in peanuts. KDR is crucial for preventing premature germination and the effects of aflatoxin contamination after maturation, ensuring peanuts reach a safe dehydration level. Arachidic acid, a key component of saturated fatty acids, has been linked to increased risk of atherosclerosis and cardiovascular disease when consumed in excess. Therefore, reducing its content is vital for human health.

In response to your suggestion, we have conducted a thorough GWAS analysis for these two phenotypes, identifying candidate genes (**lines 496-544; Supplementary Fig. 25&26**). Further, we have incorporated gene annotation, structural analysis, haplotype analysis, population-based haplotype variation correlations with phenotypes, which confirm the essential role of these candidate genes in regulating KDR and Arachidic Acid content.

Once again, thank you for your constructive feedback, which significantly enriched our manuscript.

Title & Writing Style:

The current title is too informal, and the overall writing style lacks academic rigor (e.g., the use of "etc." in the Abstract). A more precise and professional revision should be recommended.

Authors' response: Thank you for your valuable feedback. We agree that the current title and writing style could benefit from a more formal and precise approach. In the revised version, we have refined the title to reflect the academic rigor required, and we have also revised the writing style to avoid informal language such as 'etc.' in the Abstract. We hope these changes align better with the expectations for academic writing.

In view of above, we believe that Editor and Reviewers are happy with our responses and revision of the MS.